**Global change in streamflow extremes under climate change over the 21st**
**century**
**Behzad Asadieh [1],*, Nir Y. Krakauer [2]**
[1] Department of Earth and Environmental Science, University of Pennsylvania,
Philadelphia, PA, USA; basadieh@sas.upenn.edu
[2] Civil Engineering Department and NOAA-CREST, The City College of New York, City
University of New York, New York, USA; nkrakauer@ccny.cuny.edu
* *Correspondence to :* Behzad Asadieh (basadieh@sas.upenn.edu)
**Abstract**
Global warming is expected to intensify the Earth's hydrological cycle and increase flood and
drought risks. Changes over the 21st century under two warming scenarios in different
percentiles of the probability distribution of streamflow, and particularly of high and low
streamflow extremes ($95^{th}$ and $5^{th}$ percentiles) are analyzed using an ensemble of
bias-corrected global climate model (GCM) fields fed into different global hydrological
models (GHMs), to understand the changes in streamflow distribution and simultaneous
vulnerability to different types of hydrological risk in different regions. In the multi-model
mean under RCP8.5 scenario, 37% of global land areas experience increase in magnitude of
extremely high streamflow (with an average increase of 24.5%), potentially increasing the
chance of flooding in those regions. On the other hand, 43% of global land areas show a
decrease in the magnitude of extremely low streamflow (average decrease of 51.5%),
potentially increasing the chance of drought in those regions. About 10% of the global land
area is projected to face simultaneously increasing high extreme streamflow and decreasing
low extreme streamflow, reflecting potentially worsening hazard of both flood and drought;
further, these regions tend to be highly populated parts of the globe, currently holding around
30% of the world's population (over 2.1 billion people). In a world more than 4 degrees
warmer by the end of the $21^{st}$ century compared to the pre-industrial era (RCP8.5 scenario),
changes in magnitude of streamflow extremes are projected to be about twice as large as in a 2
degree warmer world (RCP2.6 scenario). Results also show that inter-GHMs uncertainty in
streamflow changes, due to representation of terrestrial hydrology, is greater than the
inter-GCMs uncertainty due to simulation of climate change. Under both forcing scenarios,
there is high model agreement for t increases in streamflow of the regions near and above the
Arctic Circle, and consequent increases in the freshwater inflow to the Arctic Ocean, while
subtropical arid areas experience reduction in streamflow.
**1. Introduction**
Floods and droughts, the natural disasters with the highest cost in human lives (Dilley, 2005;
IFRC, 2002), are projected to become more intense under anthropogenic global warming and
climate change (Dai, 2011; Dankers et al., 2013; Field, 2012; Stocker et al., 2013).
Observational records as well as global climate model (GCM) simulations both show that the
amount of water vapor in the atmosphere increases at a rate of approximately 7% per K of
increase in global mean temperature (Allen and Ingram, 2002; Held and Soden, 2006; Wentz et
al., 2007), as expected from the Clausius-Clapeyron equation conditional to stable relative
humidity (Held and Soden, 2006; Pall et al., 2006). Increased amount of atmospheric water
content is expected to intensify precipitation extremes (Allan and Soden, 2008; O'Gorman and
Schneider, 2009; Trenberth, 2011), as evidenced by both observations and GCM simulations
(Alexander et al., 2006; Asadieh and Krakauer, 2015, 2016; Kharin et al., 2013; Min et al.,
2011; O'Gorman and Schneider, 2009; Stocker et al., 2013; Toreti et al., 2013; Westra et al.,
2013), with relatively stronger impact than for mean precipitation (Asadieh and Krakauer,
2016; Lambert et al., 2008; Pall et al., 2006). Change in intensity and distribution of
precipitation events under climate change is expected to increase the intensity and frequency of
flood and drought events in many regions (Alfieri et al., 2015, 2017, Asadieh and Krakauer,
2015, 2016; Dankers et al., 2013; Ehsani et al., 2017; Field, 2012; Held and Soden, 2006; Min
et al., 2011; O'Gorman and Schneider, 2009; Stocker et al., 2013).
Average runoff projections from 3 GCMs show strong positive trend around high latitudes
and negative trend for some mid-latitude regions, by the end of the 21[st] century (Hagemann et
al., 2013). Another study of runoff projections from a larger ensemble of GCMs also confirms
such trends in runoff for the 21[st] century (Tang and Lettenmaier., 2012). Changes in runoff, and
consequently in streamflow, under current and future climate change has strong implications
for available freshwater resources (Arnell, 2004; Brekke et al., 2009; Oki and Kanae, 2006;
Stocker et al., 2013; Vörösmarty et al., 2000). Climate change is projected to decrease mean
runoff in land areas around Mediterranean and some parts of Europe, southern Africa and
central and southern America, and consequently increase water stress in those regions (Arnell,
2004). It is also projected to worsen aridity in southern Europe and the Middle East, Australia,
Southeast Asia, and large parts of Americas and Africa, in the 21st century (Dai, 2011).
Regions experiencing increase in total annual precipitation and runoff under climate
change may also face increased water stress, as a result of change in precipitation and runoff
distribution (Arnell, 2004; Asadieh and Krakauer, 2016; Oki and Kanae, 2006). Implications of
anthropogenic climate change for flood events are widely noted in the literature; However,
there are few multi-model analyses of future change in streamflow extremes at global scale
(Arnell, 2004; Dankers et al., 2013; Hirabayashi et al., 2008, 2013; Koirala et al., 2014; Schewe
et al., 2013). A study of streamflow provided by the Inter-Sectoral Impact Model
Intercomparison Project (ISI-MIP) (Warszawski et al., 2013) projects increases for the high
latitudes, eastern Africa and India, and decreases in streamflow of Mediterranean and southern
Europe as well as South America and southern parts of North America, by the end of the 21[st]
century (Schewe et al., 2013), similar to some other studies (Hagemann et al., 2013; Tang and
Lettenmaier., 2012). Another study of ISI-MIP streamflow projects increases in 30 year return
period of high flow in major parts of Siberia and some regions around Southeast Asia, and
decreases in northern and eastern Europe and some regions around western United States, by
the end of the 21[st] century (Dankers et al., 2013). Approximately two-thirds of global land area
are projected to experience positive trend in the magnitude and frequency of 30-year return
period of high flow (Dankers et al., 2013) and magnitude of 95[th] percentile of streamflow
(Koirala et al., 2014), and have shown increase in magnitude of annual-maximum daily
streamflow (Asadieh et al., 2016). The 95[th] and 5[th] percentiles of flow have been used as
indices for analysis of streamflow extremes by United States Geological Survey (USGS) (Jian
et al., 2015) and other studies (Koirala et al., 2014). Some studies have used changes in the 95[th]
percentile of flow in gridded streamflow data to study changes in flood events (Wu et al., 2012,
2014), while the 5[th] percentile of streamflow has been used to study changes in drought events
(Ellis et al., 2010; Sprague, 2005). Although changes in high and low extremes of streamflow
may not be directly interpreted as changes in flood and drought events, since the thresholds for
flood and drought damage vary according to factors such as mean climate, the magnitude of
water demand, and engineering works for water storage and transport , such changes affect the
likelihood of occurrence of those events, and can be considered a reasonable indicator of
climate impacts on large-scale flood and drought hazard respectively (Vörösmarty et al., 2000).
Accurate simulation of weather fields such as precipitation, as well as simulation of the diverse
hydrological processes that lead to streamflow generation, are major sources of uncertainty in
streamflow simulation (Giuntoli et al., 2015; Hagemann et al., 2013; Schewe et al., 2013).
Some earlier adoptions of climate model projections for flooding studies utilized single global
hydrological models (GHMs) for flow routing and streamflow simulation under the
GCM-simulated climate (Hirabayashi et al., 2013; Koirala et al., 2014). However, the process
simulation in GHMs is also a major source of uncertainty, as flow routings in different GHMs
using the same weather fields can result in markedly different flood and drought trend
predictions (Giuntoli et al., 2015; Haddeland et al., 2011; Hagemann et al., 2013).
Additionally, historical simulations of weather variables from GCMs have shown
discrepancies (biases) compared to the observations (Asadieh and Krakauer, 2015; Ehret et al.,
2012; Hempel et al., 2013; Krakauer and Fekete, 2014), which may affect the climate change
impact projections using the GCM outputs (Hagemann et al., 2011, 2013). This issue is often
solved utilizing bias correction methods, in which the mean value of the time series is adjusted
according to the observational records, while supposedly preserving the trends (Hempel et al.,
2013), as done in ISI-MIP dataset (Warszawski et al., 2013).
A study of changes in frequency of 95[th] and 10[th] percentiles of un-routed runoff in the 21[st]
century, using multiple GCMs and GHMs from ISI-MIP under RCP8.5 scenario, shows that
the number of days with flow above the historical 95[th] percentile will significantly increases in
the high latitudes and the number of days with flow below the historical 10[th] percentile will
increase significantly in Mediterranean, southern North America, and Southern Hemisphere
(Giuntoli et al., 2015). However, changes in runoff extremes do not directly correspond to
floods of large water bodies, where routed runoff (streamflow) has been widely used instead
for this purpose (Dankers et al., 2013; Hirabayashi et al., 2013; Koirala et al., 2014).
Additionally, Giuntoli et al., 2015 studies changes in frequency of streamflow extremes, and
not magnitude/intensity. Change in frequency of extremes may be studied using the historical
extreme thresholds/percentiles, which may come to occupy different points in the streamflow
probability distribution under future climate change. A study of change in 100-yr flood return
period in the last 3 decades of the 21[st] century compared to the last 3 decades of the 20[th]
century, projected by 11 GCMs under various emission scenarios, shows increased flood
frequency over the South and Southeast Asia, northern Eurasia, South America, and tropical
Africa (Hirabayashi et al., 2013). Another similar study investigated changes in 5[th] and 95[th]
percentiles of streamflow, projected by the same 11 GCMs (Koirala et al., 2014). However,
both these studies used a single river routing model for simulating streamflow using the GCM
inputs.  However, a single multi-GCM multi-GHM global analysis of projected changes in

magnitude of streamflow (routed runoff) extremes under different warming scenarios over the 21st century is not yet available.Here, we study changes in the magnitude of the 95th percentile of annual streamflow (P95) in 21C compared to 20C, in which an increase may indicate a greater potential for flood events. We also study the change in the magnitude of the 5th percentile (P5), in which a decrease may indicate greater potential for drought events. We study changes in both extremes to understand the changes in streamflow distribution and simultaneous vulnerability profiles to different types of hydrological risk in different regions. We use daily streamflow simulations from 25 GCM-GHM combinations (5 bias-corrected GCMs and 5 GHMs) from the ISI-MIP. We analyze simulated streamflow at the end of the 21st century (2070-2099, 21C) in comparison with the end of the 20th century (1971-2000, 20C). GHM-generated streamflow based on GCM inputs does not well capture the interannual variability in flow compared to observations, even where, as in ISI-MIP, the GCM outputs are bias-corrected. However, the multi-decade average of bias-corrected ISI-MIP streamflow is shown to be similar to that of observation-based streamflow simulations (Asadieh et al., 2016). Other studies have also used relative changes in multi-decade average of streamflow percentiles in a future 21C time window compared to a historical 20C time window for flooding and streamflow extremes analyses (Dankers et al., 2013; Hirabayashi et al., 2013; Koirala et al., 2014; Tang and Lettenmaier., 2012). Alongside the study of the magnitude of change, we also study the percentage of global population affected by changes in high and low streamflow extremes, as an indication of the potential impact of changes in flood or drought events in those regions. Limiting global warming to 2 degrees Celsius above the pre-industrial era (achievable in RCP2.6 scenario (Moss et al., 2010; Stocker et al., 2013)) has been targeted in many scientific and governmental plans, for instance the 2015 Paris Climate Agreement (UNFCCC, 2015). However, the increasing trajectory of emissions observed over the beginning on the 21st century, if continued, is more consistent with around 4 degrees Celsius of warming by the end of the century (similar to RCP 8.5 scenario (Moss et al., 2010; Stocker et al., 2013)). Hence, we study both low and high radiative forcing scenarios (RCP2.6 and RCP8.5) to investigate the impacts of 21C anthropogenic forcing on streamflow extremes.

## 2. Materials and Methods

We use daily streamflow data obtained from the first phase of the ISI-MIP (Warszawski et al., 2013). The ISI-MIP streamflow projections are produced by multiple GHMs, based on bias-corrected meteorological outputs of 5 GCMs from the fifth version of the Coupled Model Intercomparison Project (CMIP5) (Dankers et al., 2013), which are downscaled to 0.5 degree

resolution for the period 1971-2099. The GCMs contributing to the first phase of ISI-MIP are: GFDL-ESM2M, HadGEM2-ES, IPSL-CM5A-LR, MIROC-ESM-CHEM and NorESM1-M (Warszawski et al., 2013). The 5 GHMs selected for this study are WBM, MacPDM, PCR-GLOBWB, DBH and LPJmL (refer to supplementary materials for details). These models which have been used in previous studies, along with other models (Schewe et al., 2013). However, we limit the number of GHMs to 5 so the analysis in this global scale is practical.

Increasing/decreasing extreme high/low streamflow can form four combinations, which are categorized as the following four quadrants: 1. Increased high extreme and decreased low extreme, 2. Increased high and low extreme, 3. Decreased high and low extreme, and 4. Decreased high extreme and increased low extreme. Results obtained are averaged for each of these quadrants and the comparison of results between different scenarios is made for each quadrant individually. Assignment of each grid cell to the specified quadrant is based on the averaged change across GCMs and GHMs.

In order to calculate the normalized change in high extreme of a grid cell, the magnitude of the 95[th] percentile of daily streamflow (P95) is calculated for each year, and then averaged for 20C (called $Q_{20C}$) and 21C (called $Q_{21C}$). The normalized change is calculated as:

$$\Delta Q = \frac{Q_{21C} - Q_{20C}}{Q_{21C} + Q_{20C}} \qquad \text{Eq.1}$$

The $\Delta Q$ value ranges between -1 and +1, where a normalized change equal to -1 indicates total loss of the 20C flow in the 21C and a normalized change equal to +1 indicates that all of the 21C flow is resultant of the change and the flow in 20C was zero. As mentioned in the Introduction, an increase in P95 suggests the potential for an increase in flooding hazards. For normalized change in low extreme of a grid cell, the same calculations are performed on the magnitude of the 5[th] percentile of annual streamflow (P5). A decrease in P5 indicates the potential for worse drought hazards, and hence, the $\Delta Q$ for P5 is multiplied by -1 when shown in the plots, so that a positive value corresponds directly to increase in potential for hydrological drought. Multi-model ensemble averages of changes are calculated based on the normalized change values. However, averaged normalized changes are then reverted to relative changes, and results are shown in both normalized change and relative percentages (cf. Figure S1). Normalized change is symmetrical with respect to zero, meaning that multiplying flow by a factor of *m* and dividing flow by *m* over the 21C both yield normalized change values with same magnitude but opposite sign. For instance, tripling the flow over the 21C will yield a

normalized change of 0.5, while dividing flow by 3 yields a normalized change value of -0.5.
Relative changes in streamflow can be very large for individual grid cells, particularly in high
latitudes that are currently ice-covered. This biases the averaging across models and grid cells
towards a positive value, as the decreases are limited to 100% loss of the historic flow, while
the increase can be well over 100% of the historic flow. Normalizing changes to between -1
and +1 is adopted here so the ranges of increases and decreases are comparable.We exclude
grid cells that have average daily flow below 0.01 mm over the period of 1971-2000
(Hirabayashi et al., 2013). Greenland and Antarctica are also excluded from the analysis. The
remaining grid cells cover 75.9% of global land area, but include 95.9% of global population as
of the year 2015. The grid cells with very low streamflow volume are excluded from the
calculations, because such regions are very sensitive to changes projected by models and small
increases in streamflow result in large relative changes in flood index, which may not
meaningfully indicate to flooding risk for such dry regions. To identify the dry grid cells, the
streamflow simulation of the WBM-plus model driven by reanalysis climate fields of WATCH
Forcing Data (WFD) is used (Asadieh et al., 2016), as the ISI-MIP uses the WFD dataset for
bias-correction of the GCM output (Hempel et al., 2013).
Calculation of normalized change in streamflow in 21C compared to 20C is performed on
each of the 25 GCM-GHM combination datasets individually. The results are averaged over
the models for each grid cell. The multi-model averages are then averaged over the grid cells
that show increase in the indicator and also separately over the grid cells that show decreases in
the indicator (two separate values for each indicator). The multi-model averages are also
averaged for each quadrant. This averaging gives a better sense of the projected magnitudes of
changes in the high and low streamflow extremes for each warming scenario in affected
regions than averaging over all land areas, because the positive and negative trends cancel each
other out in a global averaging due to the semi-symmetric behavior of changes (Figures 2.c and
d). In a supplementary analysis, the streamflow data of all the model combinations were
averaged first and the normalized change was calculated on the multimodel-averaged
streamflow data. Both approaches yielded very similar results, indicating that the analyses are
not sensitive to the method of averaging.
The two-sample t-test (Snedecor and Cochran, 1989) is used in this study to quantify the
statistical significance level of difference between the means of the 20C and 21C streamflow
time series (refer to supplementary materials). The percentage of land area with statistically
significant change (at 95% confidence level) is reported. The affected population is
calculated using the Gridded Population of the World (GPW) data from the Center for
International Earth Science Information Network (CIESIN) (Doxsey-Whitfield et al., 2015).
**3. Results and Discussion**
Based on multi-model mean results under RCP8.5 scenario, 36.7% of global land area shows
an increase in high extreme (95[th] percentile) of streamflow (whose magnitude averages
24.55%), potentially increasing the chance of flooding in those regions, and 39.2% of land area
shows an average 21.10% decrease in P95. On the other hand, 43.2% of global land area shows
an average 51.40% decrease in low extreme (5[th] percentile), potentially increasing the chance
of drought in those regions, and 32.7% of land area shows an average 30.30% decrease in P5
(Table 1). Compared to RCP8.5, RCP2.6 shows a higher percentage of land area with
increasing P95, a lower percentage with decreasing P5, and much smaller magnitudes of mean
changes (Table 1).
Figure 1 shows global maps of normalized change in median, P5, and P95 of streamflow in
21C compared to 20C under two different warming scenarios, obtained from the ensemble
mean of all 25 GCM-GHM combination datasets. Under RCP8.5 scenario, the high latitudes
show an increase in all percentiles of flow, while the Mediterranean shores, Middle East,
southern North America and the Southern Hemisphere show a decrease in all percentiles. The
United Kingdom, some parts of Indonesia, India and southern Asia show an increase in the
magnitude of P95 while experiencing a decrease in the magnitude of P5. Median flow shows a
general pattern of change similar to P5. As shown in the figure, changes are more intense in
RCP8.5 scenario (representative of 4 degrees warmer world in 21C compared to pre-industrial
era) than in RCP2.6 scenario (representative of 2 degrees warmer world in 21C compared to
pre-industrial era). However, unlike the RCP8.5 scenario, the RCP2.6 scenario projects
increase in P95 for eastern United States as well as southern and western Europe. Global maps
of change in median, P5, and P95 of streamflow for each individual model, are shown in
supplemental Figures S2-7.
Figure 2 depicts the multi-model mean changes in high and low extremes of streamflow
averaged by latitude, as well as the scatter of the grid cells over the defined quadrants, under
each RCP scenario. Results show increasing P95 (and thus increased potential for flooding)
and increasing P5 (and thus decreasing potential for drought) in high latitudes, especially in the
regions near and above the Arctic Circle, in both warming scenarios. The changes are projected
with high agreement among the models in both scenarios, with greater change in RCP8.5
compared to RCP2.6 (Figure 2). This indicates future increase in the flow volume of the Arctic
rivers and increased freshwater inflow into the Arctic Ocean, continuing the trend observed
over the last decades (Peterson et al., 2002; Rawlins et al., 2010), which can be attributed to the
thaw of permafrost and increased precipitation in a warmer climate. Rivers play a critical role
in the Arctic freshwater system (Carmack et al., 2016; Lique et al., 2016), as river runoff is the
major component of freshwater flux into the Arctic Ocean (Carmack et al., 2016). Arctic
rivers' inflow to the Arctic Ocean accounts for around 10% of global annual water flux into the
oceans (Haine et al., 2015; Lique et al., 2016). The projected increase in meltwater flux into the
Arctic Ocean may contribute to sea level rise and changes in water salinity, temperature as well
as circulation in the Arctic Ocean (Peterson et al., 2002; Rawlins et al., 2010). The Southern
Hemisphere shows a general decreasing trend in both P5 and P95, indicating a negative trend in
flow volume. The Northern Hemisphere tropics however show a mixed trend, as changes
averaged over latitude show fluctuations between latitudes within the tropics (Figure 2).
Figures 3 and 4 depict multi-model changes in streamflow extremes under different
warming scenarios, averaged over different latitudinal windows. Figure 3 shows the results
from streamflow routings of each GHM based on inputs from multiple GCM simulations,
where the thick lines in the plots denote the mean of change in the indicator and the shades
denote ±1 st. dev. For each single GHM (shown by distinct colors), the thick line in the plots
show the average of GCMs and the shading denotes the standard deviation of GCMs. Hence,
the shadings in this figure are representative of uncertainties arising from GCMs. Also,
different average values (thick lines) means that different GHMs have produced different
streamflow routings and different change values in the indicators, even though the routings are
based on inputs from the same ensemble of GCMs. Figure 4, on the other hand, shows
streamflow routings of multiple GHMs based on inputs from each of the GCMs, where the
thick lines in the plots denote the mean of change in the indicator and the shades denote ±1 st.
dev. For each single GCM (shown by distinct colors), the shading denotes the standard
deviation of GHMs and hence, is representative of uncertainties arising from GHMs. The
RCP8.5 scenario show higher normalized change values and larger uncertainties, compared to
the RCP2.6 scenario. The uncertainties are proportionally greater for P5 trend projection than
for P95 (Figure 3 and 4).
The shadings in Figure 4 (inter-GHM uncertainty) is broader than in Figure 3 (inter-GCM
uncertainty), which shows that the GHMs contribute to higher rate of uncertainties in
streamflow change projections than GCMs. As seen in Figure 3 (c-d), for instance, the P5
predictions of the DBH hydrological model for Northern Hemisphere are significantly
different from the other 4 hydrological models considered here, even though the streamflow
routings are based on the same GCM inputs. Such inconsistency between DBH models and
other models' results may not be detectable, if, as in the Figure 4, only the mean and standard
deviation across GHMs is shown. High uncertainties in Northern Hemisphere low extreme
trends in Figure 4 (c-d) reflects large disagreements among the GHMs for that region, while
Figure 3 (c-d) reveals the major cause of such uncertainties to be the DBH model.
Figure 5 illustrates the global maps of combined change in high and low streamflow
extremes under each RCP scenarios, obtained from the multi-model mean results across all 25
GCM-GHM combination datasets. Grid cells falling in each of the defined quadrants are
shown with different colors, saturation of which is representative of the intensity of changes.
As shown in the Figure, northern high latitudes, especially north Eurasia, northern Canada and
Alaska, as well as eastern Africa and parts of South and Southeast Asia and Eastern Oceania
show increase in the magnitude of high streamflow extremes (P95) in both scenarios, similar to
findings of earlier studies and reflecting a potential for increasing flood hazard (Dankers et al.,
2013; Hirabayashi et al., 2013; Schewe et al., 2013). Central America, Southern Africa, Middle
East, Southern Europe, Mediterranean and major parts of South America and Australia show
decrease in the magnitude of low streamflow extrem (P5) in both scenarios, comparable to
findings of earlier studies and reflecting a potential for increasing drought hazard (Arnell,
2004; Dai, 2011; Hagemann et al., 2013; Schewe et al., 2013). The United Kingdom and the
shores of the North Sea as well as large parts of Tibetan, South Asia and Western Oceania show
increase in potential for both flood and drought hazards (increase in P95 and decrease in P5). In
these cases, while preserving the direction of change, the RCP8.5 scenario projects stronger
magnitude change compared to the RCP2.6 scenario. Southern and Western Europe and
southern parts of the United States show small-magnitude, mixed-sign changes in P95 and P5
in the RCP2.6 scenario. However, projections under RCP8.5 scenario are for strong decrease in
P5 in those regions, suggesting increasing potential for drought hazard. Some parts of eastern
Russia and northern United States show decreases in P95 and increases in P5, suggesting the
potential for reduction in both flood and drought hazards (Figure 5).
Under the low radiative forcing scenario (RCP2.6), 45.4% of global land area shows
increase in high extreme in the multi-model mean and 36.4% shows decrease in low extreme,
indicating more land area exposed to increasing flood hazard compared to to drought hazard.
The high radiative forcing scenario (RCP8.5) projections show the opposite outcome, with
increased high extreme streamflow in 36.6% of global land area and decreased low extreme in
43.2%. Unlike the RCP2.6 scenario, the RCP8.5 scenario projects more land area exposed to
increasing drought hazard compared to flood. Moreover, changes in streamflow extremes are
larger in magnitude in RCP8.5 compared to RCP2.6, as the relative change values for 21C are
approximately double; for instance, comparing the relative increases in high extreme in Quad.2
(30.2% vs. 15.1%), and relative decrease in low extreme in Quad.3 (62.2% vs. 28.1%) (Table
3). Under RCP8.5 scenario, change in high and low extremes in 54.0 and 64.9%, respectively,
of the global land area is statistically significant. The significance fraction is lower for the
RCP2.6 scenario (38.4 and 53.8% of global land area in high and low, respectively). The
significance percentage is calculated for the multimodel-averaged streamflow time series in
21C compared to 20C, and the percentages for each individual model may be different.
Under RCP8.5 scenario (and similarly in RCP2.6), nearly 9.6% of global land areas show
increasing potential exposure to both increase flood and drought hazards (increasing P95
combined with decreasing P5). Unfortunately, these regions are dominantly highly populated
parts of the globe, the residence of around 29.6% of the world's current population, or more
than 2.1 billion people (Table 2). The 2015 Paris Climate Agreement, adopted at the 21[st]
meeting of the Conference of Parties (COP21), targets to limit the global temperature rise "well
below" 2°C above the pre-industrial levels (UNFCCC, 2015). Even though seeming to be
ambitious, such an agreement in intergovernmental level is a start to motivate the developed
countries producing the majority of greenhouse gases to limit emissions and finance the
climate-resilient development in lower income economies, and, based on the projections
analyzed here, would limit changes in streamflow extremes that correspond to the potential for
increasing flood and drought hazards in many densely populated areas.
**4. Conclusion**
Global daily streamflow simulations of 25 GCM-GHM combination datasets are analyzed to
study the implications of increased GHG emissions and consequent atmospheric temperature
rise for global streamflow extremes. The projected changes in high and low streamflow
percentiles in the 21C compared to the 20C were studied, under both low and high radiative
forcing scenarios, to investigate the changes in streamflow distribution and simultaneous
vulnerability to different types of hydrological risk in different regions, and study the number
of people affected by such changes. Multiple GHMs and GCMs are used to account for
uncertainties arising from the hydrological models and flow routing process on the flood and
drought studies, additional to the weather field simulation uncertainties.
Results suggest that northern high latitudes, especially north Eurasia, northern Canada and
Alaska, as well as Tibet Plateau and southern India will face strong increases in high extreme
of streamflow over the 21st century, with the potential for increasing flood hazard in those
regions. The Mediterranean shores, Middle East, southern North America and the Southern
Hemisphere are projected to see strong decrease in low extreme of streamflow, with the
potential for increasing drought hazard for those areas. The projected increase in meltwater
flux from the pan-Arctic watershed into the Arctic Ocean may contribute to sea level rise, and
changes in salinity, temperature and circulation in the Arctic Ocean. The United Kingdom and
the shores of the North Sea as well as large parts of Tibetan, South Asia and Western Oceania
show increase in potential for both flood and drought hazards. Regions projected to experience
simultaneous increases in both flood and drought chances as a result of change in streamflow
distribution, are highly populated parts of the globe, even though covering a small fraction of
global land area. A world 2°C warmer than the pre-industrial era will still face increases in
flood and drought in most regions. However, the GCM and GHM ensemble projects that 4°C
of warming will bring nearly twice as much increase in the magnitude of high and low
streamflow extremes that, in many densely populated areas, are likely to correspond to
high-impact flood and droughts.
Similar to previous studies (Giuntoli et al., 2015; Hagemann et al., 2013), our results show
that GHMs contribute to more uncertainty in streamflow changes than the GCMs, where
different GHMs have produced different streamflow routings and different change values in
the extremes, even though the routings are based on inputs from the same ensemble of GCMs.
Our findings suggest that in addition to inclusion of ensembles of GCMs for hydrological
impact assessments in lieu of a single model, inclusion of ensembles of GHMs, as done in
projects like ISI-MIP, may further improve accuracy of projections. The bias-correction
applied on GCM outputs in ISI-MIP may help reduce the uncertainties of climate models in
hydrological impact assessments. However, high inter-GHM uncertainties suggest that more
focus is needed on improving the process representation and calibration of hydrological
models, so that the next generations of climate-hydrological model intercomparison projects
yield higher agreement on future hydrological hazard assessments.
**Author Contribution**
B. Asadieh and N.Y. Krakauer conceived and designed the experiment. B. Asadieh carried out
the analyses and wrote the draft manuscript. B. Asadieh and N.Y. Krakauer analyzed the
results, and wrote the manuscript.
**Acknowledgments**
The authors gratefully acknowledge support from NOAA under grants
NA11SEC4810004, NA12OAR4310084, NA15OAR4310080 and NA16SEC4810008, and
from PSC-CUNY Award # 68346-00 46. All statements made are the views of the authors and
not the opinions of the funding agency or the U.S. government.
**Conflict of interest**
The authors declare that they have no conflict of interest.

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

Table 1. Multi-model average change in high and low streamflow extremes, as well as
percent of population and land area affected by each category, for RCP2.6 and
RCP8.5 scenarios. Presented percentages are for total global land area and total global
population, and sum up to the 75.9% of global land area and 95.9% of the year 2015
total global population considered in this study. The value of change for indicators are
normalized change and the numbers in parenthesis show the changes reverted to the
relative percentages.

| | Normalized (and Percent) of Change in magnitude of extremes | | Land Area Affected (% of total 148.9 million km$^2$, sum up to 75.9%) | | Population Affected (% of total 7.13 billion people, sum up to 95.9%) | |
|---|---|---|---|---|---|---|
| | RCP 8.5 | RCP 2.6 | RCP 8.5 | RCP 2.6 | RCP 8.5 | RCP 2.6 |
| High extreme (P95) Increased cells (*Increased flood potential*) | 0.1093 (24.55 %) | 0.0606 (12.90 %) | 36.7% | 45.4% | 53.7% | 62.7% |
| High extreme (P95) Decreased cells (*Decreased flood potential*) | -0.1178 (-21.10 %) | -0.0539 (-10.25 %) | 39.2% | 30.5% | 42.2% | 32.2% |
| Low extreme (P5) Decreased cells (*Increased drought potential*) | -0.2045 (-51.40 %) | -0.1029 (-22.95 %) | 43.2% | 36.3% | 67.8% | 56.1% |
| Low extreme (P5) Increased cells (*Decreased drought potential*) | 0.1784 (30.30 %) | 0.1018 (18.50 %) | 32.7% | 39.6% | 28.1% | 39.8% |

Table 2. Percent of population and land area affected by each high and low extreme change quadrants, for RCP2.6 and RCP8.5 scenarios. Presented percentages are for total global land area and total global population. Hence, the percentages presented for quads. 1-4 sum up to the 75.9% of global land area and 95.9% of the year 2015 total global population considered in this study.

| | | Quad. 1. increased high extreme and decreased low extreme | Quad. 2. increased high and low extreme | Quad. 3. decreased high and low extreme | Quad. 4. decreased high extreme and increased low extreme |
|---|---|---|---|---|---|
| Land area affected (% of total 148.9 million km$^2$) | RCP8.5 | 9.6% | 27.0% | 33.6% | 5.7% |
| | RCP2.6 | 10.8% | 34.5% | 25.5% | 5.1% |
| Population affected (% of total 7.13 billion people) | RCP8.5 | 29.6% | 24.1% | 38.2% | 4.0% |
| | RCP2.6 | 27.1% | 35.6% | 28.9% | 4.3% |

2
3

Table 3. Multi-model average change in high and low streamflow extremes, averaged for each quadrant, for RCP2.6 and RCP8.5 scenarios. The numbers show the normalized change and the numbers in parenthesis show the changes reverted to the relative percentages.

| | Quad. 1. increased high extreme and decreased low extreme | | Quad. 2. increased high and low extreme | | Quad. 3. decreased high and low extreme | | Quad. 4. decreased high extreme and increased low extreme | |
|---|---|---|---|---|---|---|---|---|
| | Change in high ext. | Change in low ext. | Change in high ext. | Change in low ext. | Change in high ext. | Change in low ext. | Change in high ext. | Change in low ext. |
| RCP8.5 | 0.0481 (10.10 %) | -0.0901 (-19.80 %) | 0.1311 (30.20 %) | 0.1909 (32.05 %) | -0.1290 (-22.85 %) | -0.2372 (-62.20 %) | -0.0508 (-9.65 %) | 0.1183 (21.15 %) |
| RCP2.6 | 0.0306 (6.30 %) | 0.0556 (-11.80 %) | 0.0700 (15.05 %) | 0.1074 (19.40 %) | -0.0593 (-11.20 %) | -0.1230 (-28.05 %) | -0.0267 (-5.20 %) | 0.0635 (11.95 %) |

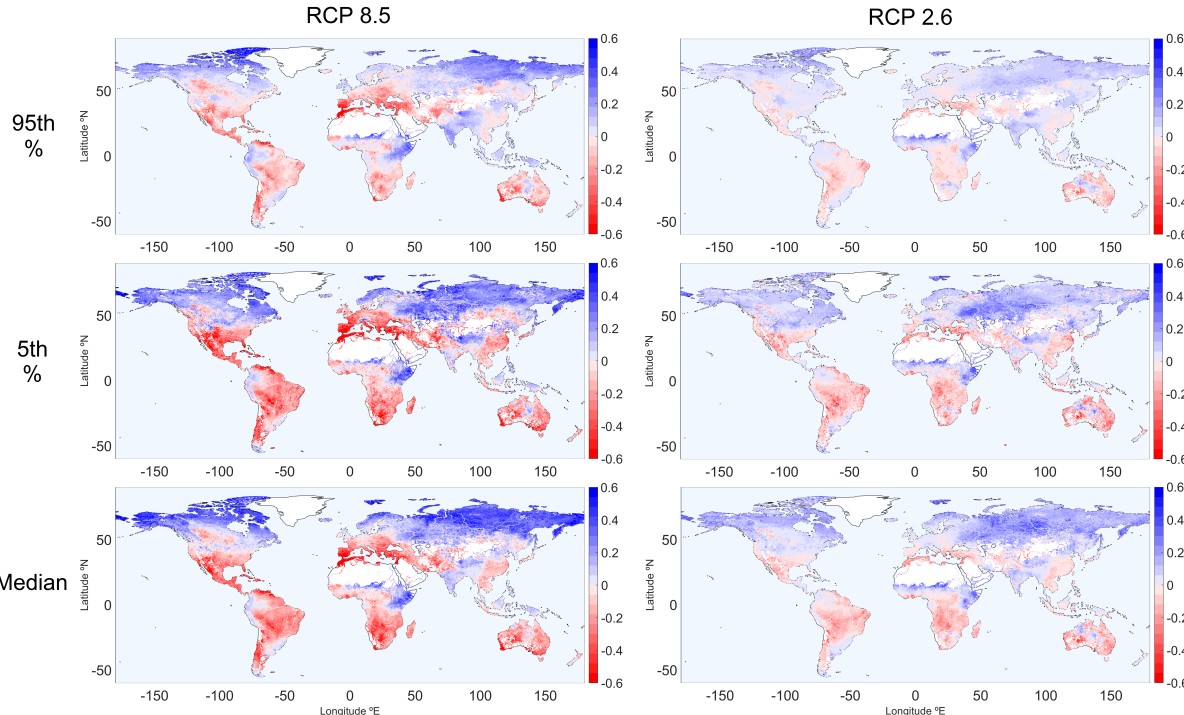

Figure 1. Global maps of normalized change in different streamflow percentiles (95th,
5th and median), under the RCP8.5 and RCP2.6 scenarios. Maps show the ensemble
mean results of all 25 models.

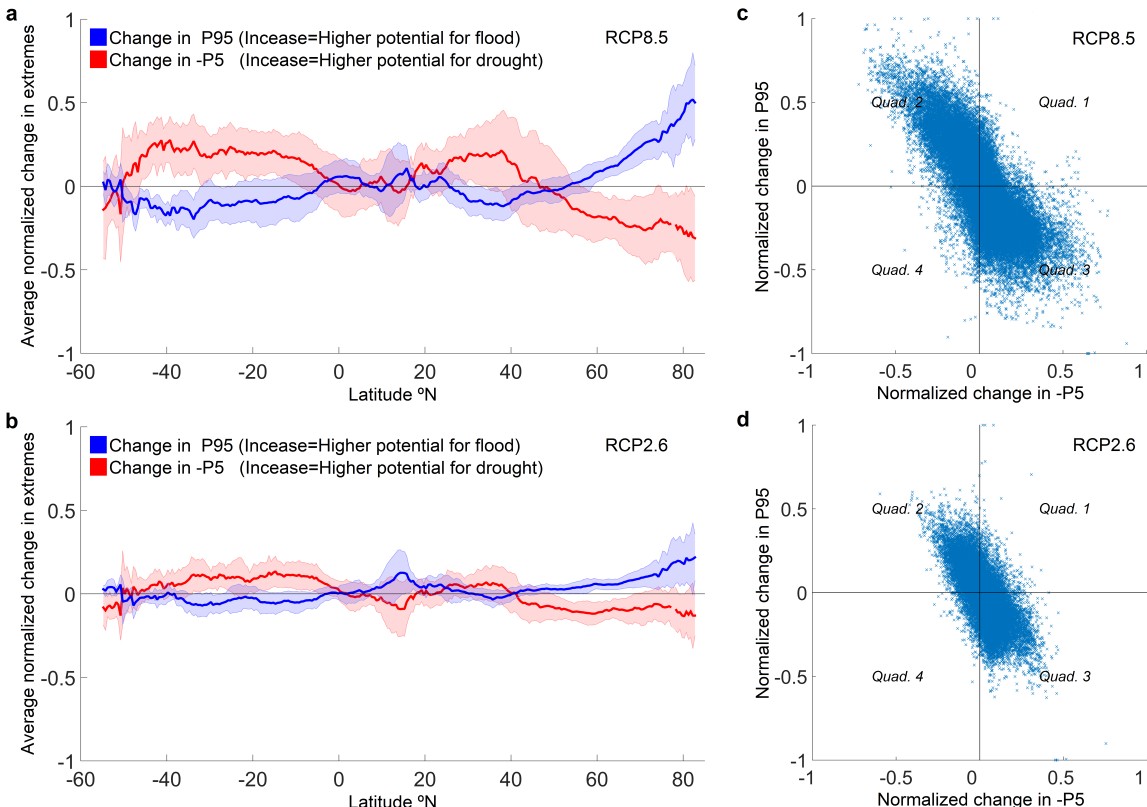

Figure 2. Multi-model change in P95 and P5*-1 under (a) RCP8.5 and (b) RCP2.6 scenarios, averaged by latitude, and scatter plot of change for each grid cell under (c) RCP8.5 and (d) RCP2.6 scenarios. The thick line in the panels a and b show the ensemble mean value of all 25 GCM-GHM combination datasets and the shading denotes ±1 st. dev.

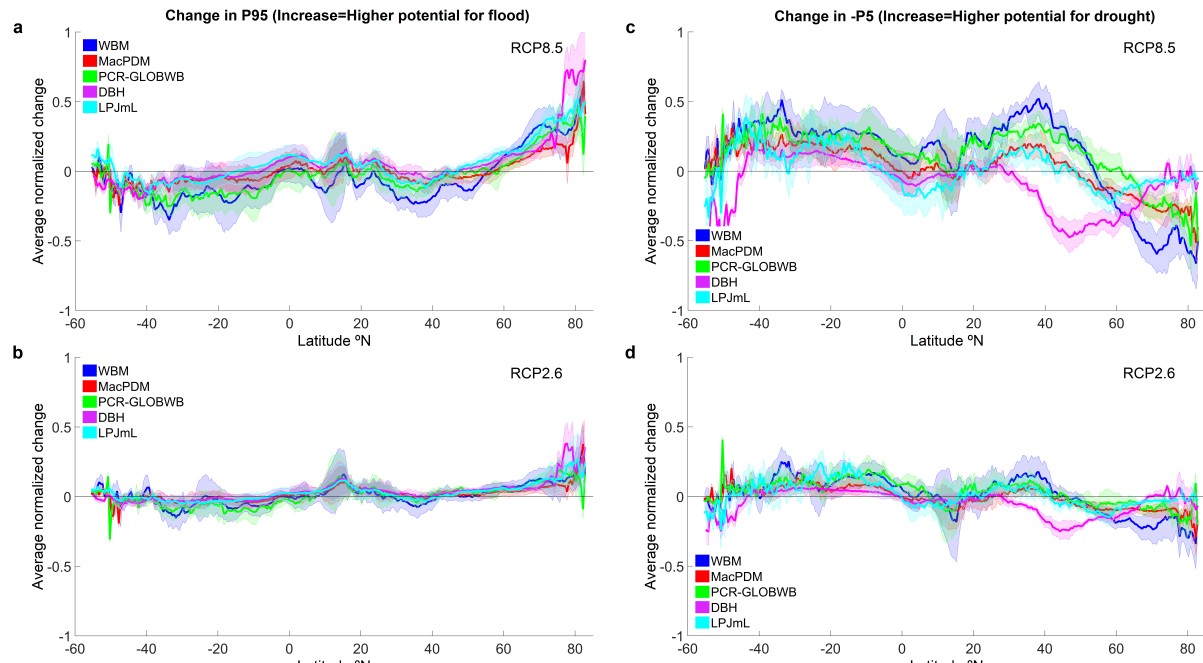

Figure 3. Multi-model change in P95 under RCP8.5 (a) and RCP2.6 (b) scenarios, and change in P5*-1 under RCP8.5 (c) and RCP2.6 (d) scenarios, averaged by latitude. The thick lines in the plots show the mean change in the indicator, based on the streamflow routings of each GHM based on inputs from multiple GCMs, and the shades denote ±1 st. dev.

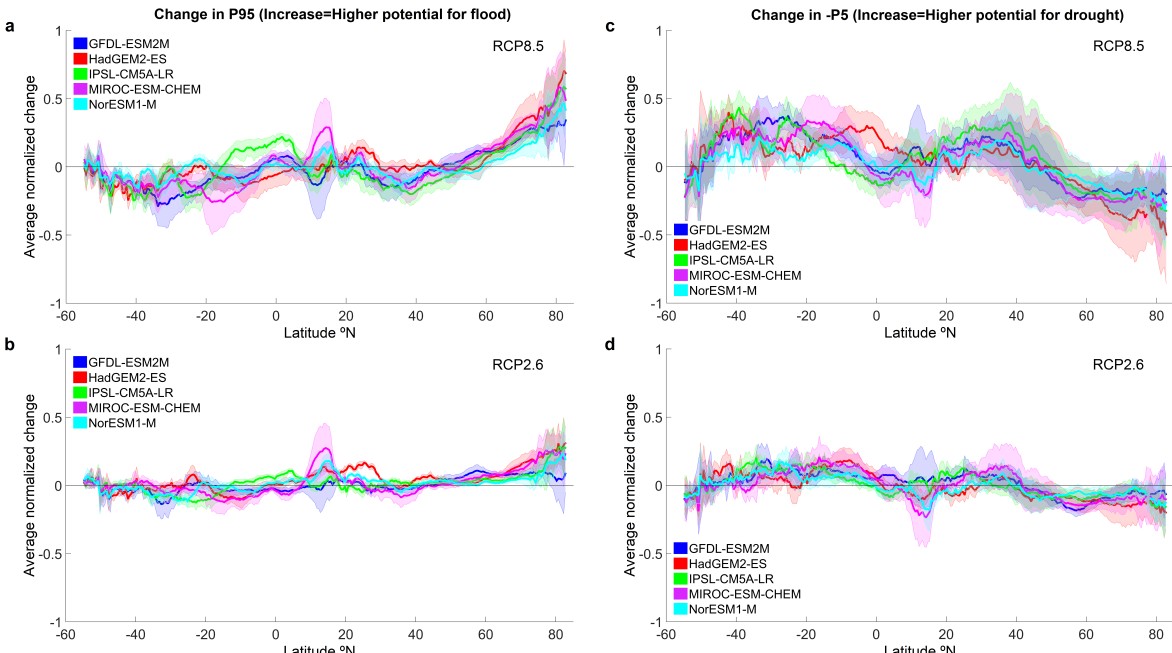

Figure 4. Multi-model change in P95 under RCP8.5 (a) and RCP2.6 scenarios (b), and change in P5*-1 under RCP8.5 (c) and RCP2.6 scenarios (d), averaged by latitude. The thick lines in the plots show the mean change in the indicator, based on the streamflow from each GCM's simulated climate routed by multiple GHMs, and the shading denotes ±1 st. dev.

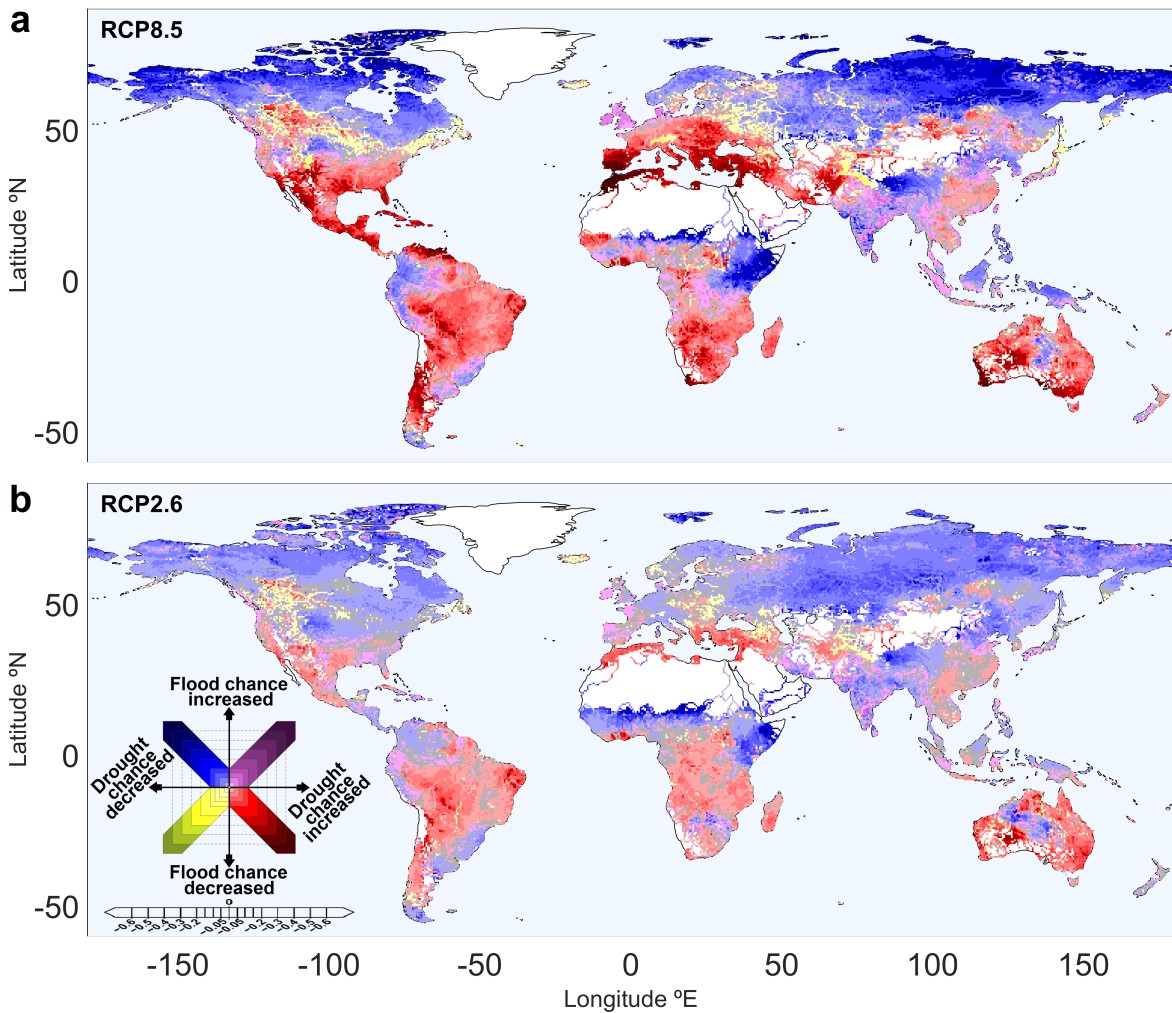

Figure 5. Global map of combined change in high and low extremes (related to change in flood and drought chance) under (a) RCP8.5 and (b) RCP2.6 scenario. The maps show the ensemble mean results of all 25 GCM-GHM combination datasets. Grid cells with increase in both flood and drought chances (Quad. 1) are shown in purple shade, cells with increased flood chance (Quad. 2) and drought chance (Quad. 3) are shown in blue and red shades, respectively, and cells with decrease in both flood and drought chances (Quad. 4) are shown in yellow shade. The saturation of colors are chosen based on the magnitude of normalized change in high and low extremes of streamflow, as shown in the legend. Distribution of cells in each of the quadrants are comparable to the Figures 2.c and d. Grid cells with normalized changes less than 1% (equal to 2% in relative terms) in each quadrant are considered as no change cells and are shown in gray.