# Peer review of "Global change in streamflow extremes under climate change over the 21st"

_Hydrology and Earth System Sciences, 2017_

## Referee Comment (RC1) · Anonymous Referee #1 · 27 Jun 2017

Review of "Global change in flood and drought intensities under climate change in the 21st century".

This paper aims at understanding how climate change will manifest in changes in flood and drought conditions globally. To pursue this, the authors collected ISI-MIP streamflow projections (which are based on bias corrected output of five global climate models (CMIP5), and five hydrological models) at a 0.5x0.5 spatial scale. Climate scenario's that are explored include RCP8.5 (which assumes emissions continue to rise throughout the 21st century) and RCP2.6 (which assumes emissions peak between 2010-2020, with emissions declining substantially thereafter).

[Figure]

To quantify changes in flood and drought conditions, the authors calculated the "normalized change" which they defined as the difference between mean percentiles (5th for low flows and 95th or high flows) of 20th century flow conditions compared to 21st century flow (normalized by the sum of 20th and 21st flow percentiles). Calculations are performed for areas where streamflow >0.01 mm/d, and exclude Greenland. Global maps of changes are provided.

The results suggest that: (i) Globally, both floods and drought are expected to intensify. (ii) in some regions (especially several highly-populated areas), both flood and drought intensity will intensify (iii) Especially, in northern high latitudes flood intensities will increase. (iv) RCP8.5 leads to nearly 2 times higher impact on changes in flood and drought intensities compared to RCP2.6. (v) Hydrological models have a larger contribution to uncertainty in the projections, compared to the effect of different climate models.

General comments

Understanding the nature of future floods and drought, and the effect of changing climate conditions on these hydrological extremes is very relevant for HESS. The analysis is rather straightforward and reasonably well-explained.

However, I do have several (major) concerns that need to be addressed before I recommend publication of this manuscript in HESS:

1) What makes the metric you define to characterize changes in flood and low intensities appropriate for the problem that you address? While the 5th and 95th flow percentiles are surely representing higher and lower flow conditions, they are not real hydrologic extremes. For example, if the 95th percentile represents flood conditions, every grid cell would experience ~18 flood days per year. Would it not be much more useful to quantify changes in the more extreme conditions (e.g. 99th percentile?). It seems that this choice arises from Giuntoli el al (2015) (except that they use the 10th percentile for low flow), but does that warrant that it actually is an appropriate metric to

use?

2) A lot of literature is already available on the topic of projected flood and drought (/low flow) changes. While you cite several papers in the introduction, it remains unclear to me which knowledge gap your paper fills compared to earlier work. I made a list of detailed suggestions in the comments below, which may help to address this issue. However, right now when I read the abstract I do not see understand knowledge gap you address, when I read the introduction no I get some idea of what has been done before but still no clear niche is identified that you fill. Any discussion is a description of the results rather than how we better understand global flood and drought changes, and in the conclusions, you summarize findings, but still I have difficulty how this relates to the vast amount of literature on previous flood changes (also what's new compared to Arnell et al. 2016, Alfieri et al. 2015, 2017, Giuntoli el al. 2015?). While I do not doubt that are new things in your paper, they need to be clearly identified.

3) Related to the previous points: the rationale behind some choices in the analysis is missing. Can you answer: (i) why we would be interested in flood and drought changes simultaneously. (ii) why the metric you choose are appropriate to characterize what you do (iii) why correlating population density and change in flood hazard is meaningful. (iv) why looking at "intensity" is novel and important compared to "frequency (which is already available using a similar approach). (v) why do we also need to look at changes in median flow (when the purpose seems extreme flows)

I do not try to imply that these choices are not well thought out and relevant. However, I do think you need to take the reader by the hand in why such choices are made

4) 0.5 x 0.5 degrees and daily forcing seems like very large spatial and temporal scales to resolve the hydrology of flood processes. I understand it is "the best you can work with" right now if you want to understand flood changes globally for the entire 21st century. However, can you better reflect on how this actually affect the degree to which you can resolve flood changes? Many different flood generating processes cannot be

represented at these scales. For example, flash floods (e.g. I expect you need sub-daily P for this?) or snowmelt driven floods (e.g. I expect that you need to parametrize the sub-grid heterogeneity of snow conditions) seem challenging, while in many places these processes are very important (e.g. see Berghuijs et al., 2016). Since there essentially is very little science in the paper (basically there are no rejectable hypotheses you test, and the paper provides a summary of available data), I think you need to say a few useful things on this topic, such that it still meets the standards of a journal like HESS.

5) I suggest to be careful with the using low flows and drought interchangeably. Low flows and droughts are not the same. You quantify changes in low flow conditions, not in drought conditions (which reflect some deviation compared to the normal flow conditions of a catchment during a particular time of the year). These two different concepts should not be mixed. (e.g. see Van Loon, 2015). To some degree the same applies for high flows and floods. However, this difference seems less important, since those two concepts are more closely related.

6) The changes in flood and drought indices per latitude intrigue me (Fig 2). How can they virtually be the inverse of another (when organized per latitude)? Is this the physical reality or an artifact of how this study quantifies changes in these aspects? Surely, floods and drought within one area can be connected (since they are part of the same climate and landscape system). However, (in my opinion) this almost one-on-one inverse pattern seems to require some attention.

7) You quantify the percentage of land area that undergoes significant streamflow change. However, the two-sample t-test that you use for this (as explained in the supplementary material) seems inappropriately used. Why would scaling this relationship by the streamflow time series variances be appropriate here? Should they not be scaled by some measure of (e.g. between year) variability in the hydrologic extremes flow?
8) By presenting only mean values of results (e.g. the mean magnitude of flood and drought changes) the reader has no idea from what distribution of changes these mean values are derived. This seems rather basic information that can easily be presented in will provide valuable insight for the reader. Right now, some of the numbers you use in key points of your paper are difficult to interpret because we have no idea if they represent many small changes and a few extreme ones, or because they represent consist medium size changes.

9) Please address the list of comments I provide below. (Several of these comments go beyond small technical details.)

detailled comments

Page 1 Line 11-12. Consistent with my main comments above, it would be very useful to have a transition sentence that actually introduces the knowledge gap in flood and drought risk projections that the paper aims to fill. Preferable reflect on that goal in the end of the abstract.

Line 16: without a definition of what aspects of these hydrological extremes you actually look at (e.g. duration, or magnitude, or both)) these precise percentages not very useful. Being more specific in line 12 may resolve this issue. The same problem applies to all other percentages provided in the abstract. (or the statement in lines 22-24)

Line 17-19: "the averaged rates of increase" (can) suggest that you exclude places where it reduced? Or is this the average of all increases and decreases globally?

Line 19: "potential risk" or "are projected" (since I guess all areas are under the "potential risk"?)

Line 20: "rate" or "change" (or "increase")?

Line 21. Semi-column or just start a new sentence?

Line 26-29: It seems odd to me that the paper suddenly talks about changes in stream-flow (I guess that means mean runoff?) while the rest of the paper is about extremes?

Line 15-18: Sure: more extreme P can lead to more extreme runoff. However, there is so much more going on that dictates runoff response (e.g. antecedent moisture conditions in floods etc). Would it be worth to say one or two things about other mechanisms that underlie floods? In many places, there is a disparity between extreme rainfall and flooding, or between lowest P and lowest Q, since so many other factors are also important (e.g. seasonal moisture conditions). Emphasizing which other processes are important may help to understand the reader what the added value is of adding the GHM's to the game (since they at least theoretically should represent all these processes that go beyond extreme P). Nor can I logically connect more extreme high P to more extreme drought (without some extra information about changes in dry spells, or hydrologic partitioning)

Line 19-29: I do not see why the paper needs to talk about changes in "mean stream-flow conditions" since it distracts from what you're really interested in (which are the hydrologic extremes)

Line 31-33: Sure, that a decrease in P can decrease runoff. However (like you give with the following example) you can also think of conditions where this does not apply.

Line 1-5: Ok, I understand that there may not be many studies that use ensembles. However, still that does not answer the question of what knowledge gap you can fill with your approach. What do we not understand because we haven't run particular ensemble projections yet?)

Line 4-5: The detection of areas that are expected to experience both more floods and drought sounds interesting at first, but what is again the knowledge gap that the paper

fills compared to earlier work, and what is the merit in identifying these at the same time (there are reasons why this can be valuable, but they need to be presented to the reader).

Lines 5-17: put these findings into context of the novel thing you're going to expose/test. Right now, it reads like a random list of previously reported streamflow changes, which are unclear why they're directly relevant to the paper.

Lines 18-20: Maybe a reference (or two) can help to support this statement?

Line 20: remove "trend" (since there may not be one)

Line 13-14: Be very explicit to the reader what the difference between "frequency" and "intensity" is, and emphasize why this difference is relevant.

Line 13: "this study" may be unclear because it can refer to your own work or the work of Giuntoli

Line 15: It may be worth to start state "Here we" and then list the "goal", rather than directly go into the "methods". This will make the list of subsequent steps outlined in the rest of the introduction much more logical. For example, right now it sounds fun that you also investigate the link with human populations, but I have no idea (or at least it's up to my own guess!) why you'll be doing this.

Line 13: why are these five GHM's selected? (after line 15-17 that question still stands)

Line 18-19: do you have any references that show this, or did this only appear in your own work?

Lines 19-21: consider rewriting this sentence

Lines 18-27: it seems a bit confusing to justify normalization before you define the

metric that you adopted. ă

Line 28 – Line 2: Is "no change" also an option?

Line 3- 14: Why did you choose the 5th and 95th percentile and not "real extremes" (see earlier comment above)

Line 16: 200?

Lines 26- 29: And what about any places where there are insignificant changes? ă

Lines 30-3: why don't you use the absolute value (and then you don't need to separate by quadrant).

Line 3-7: Why would you even bother to try that method? It seems like this method is just less logical at the start (because it is very sensitive to absolute runoff changes between models), and hence should not be considered at all?

Lines 8-13: Your results suggest that 95% of the projected flood changes are significant, but what does that really mean? Does that imply that for 95% of the grid cells you are very certain about the projections? Or does it mean that model projections may show a significant change, but all other biases and uncertainties not accounted for may lead to much lower certaities of projected change?

Also (in the supplementary material), why would "streamflow time series variances" be a relevant scale of variance here (rather than something like the variance of annual maxima or Q95).

Line 16-18: you need to show the distribution of changes, rather than just the mean value. Right now I have no idea if the mean results from consist small changes, or a few very big changes.

Line 19: why bother with median flows? I thought this paper was about the extremes?

Figure 2: The changes in flood and drought indices per latitude intrigue me. How can they virtually be the inverse of another (when organized per latitude)? Is this a physical reality or an artifact of how this study quantifies changes in these aspects?

Figure 5: it is impossible to read the scale bar in the far bottom left (on a printed page).

Line 30: you were not interested in decreases?

Line 31 because people live in a grid cells where floods increase does not mean they are affected. That depends on many other factors. ÂăCorrect?

While I appreciate, you repeat all the main results of the paper, I think the paper really needs to reflect on what we learned compared to earlier work, rather than list what came out of some modeling exercises. Âă

Table 1: without information on the distribution of changes, I have no idea about what these mean values of change represent.

Reference list: what does :"(80-. )" do in several references?

References

Arnell, N. W., & Gosling, S. N. (2016). The impacts of climate change on river flood risk at the global scale. Climatic Change, 134(3), 387-401.

Alfieri, L., Burek, P., Feyen, L., & Forzieri, G. (2015). Global warming increases the frequency of river floods in Europe. Hydrology and Earth System Sciences, 19(5), 2247-2260.

Alfieri, L., Bisselink, B., Dottori, F., Naumann, G., Roo, A., Salamon, P., ... & Feyen, L.

(2017). Global projections of river flood risk in a warmer world. Earth's Future, 5(2), 171-182.

Van Loon, A. F. (2015). Hydrological drought explained. Wiley Interdisciplinary Reviews: Water, 2(4), 359-392.

Giuntoli, I., Vidal, J. P., Prudhomme, C., & Hannah, D. M. (2015). Future hydrological extremes: the uncertainty from multiple global climate and global hydrological models. Earth System Dynamics, 6(1), 267.

Berghuijs, W. R., Woods, R. A., Hutton, C. J., & Sivapalan, M. (2016). Dominant flood generating mechanisms across the United States. Geophysical Research Letters, 43(9), 4382-4390.

Trigg, M. A., Birch, C. E., Neal, J. C., Bates, P. D., Smith, A., Sampson, C. C., ... & Ward, P. J. (2016). The credibility challenge

---

## Referee Comment (RC2) · Anonymous Referee #2 · 11 Jul 2017

The article by Asadieh and Krakauer investigates the very topical issue of flood and drought changes under future climate conditions. The topic has been subject to a large number of studies in the past few years, many of them based on the same set of GCM-GHM combinations from the ISI-MIP initiative, so it is difficult to find some unexplored topic of research in this area. However, this work is based on an interesting idea of comparing together increases in droughts and flood intensity and frequency under future climate, and I think it has potential for being published. The writing style is up to international standards and the article is compact, hence I don't see room for shortening.

[Figure]

My main concern is the misleading use of the terms "floods" and "droughts" throughout the article, for indicating high and low streamflow quantiles which are not really extremes, and certainly not linked to actual flood or drought events. Floods are normally linked to much higher quantiles, and in addition, they depend on the local vulnerability. Streamflow droughts (which by the way should be specified in the article, as meteorological and agricultural droughts are calculated differently) are also not as simple as a connection to the streamflow quantile, but they depend on the duration and intensity of the droughts. My suggestion is to clarify well through the article (e.g., p4 l18-21, p5 l22-25, p6, and in general in the results) and in the title that the aim is to "high and low streamflows" rather than floods and droughts. Interestingly, only in the caption of Fig 1 did the authors write a warning about linking those streamflow quantiles to actual floods and droughts.

Specific comments

P1 l11-12: This sentence reads more like a finding rather than an introduction. I'd move it to the introduction and support it with some references.

P2 l17-18: I suggest complementing the list with the more recent studies by Alfieri et al. (2015, 2017) and Winsemius et al. (2016).

P2 l30-31: "Climate-change-induced" could be removed here, to avoid speculation.

P5 l14-15: The sentence doesn't read well. Please reformulate.

P5 l19: currently-frozen should be replaced with more appropriate terminology. Also, this sentence needs a supporting reference or a reason for the wider model spread.

P6 l28: also the over –> also over

P7 l4: Is it available? Otherwise you should add "not shown"

P8 l14: flux to the Arctic Ocean

P8 l26-27: "In the meantime" should be replaced with more appropriate terminology.

P11 l5-7: This sentence sounds speculative as no specific simulation was performed to support it.

Table 1: I suggest removing "rel" in the first two columns, as that is clear from the % sign.

Figure 5 is surely the most interesting one, and the main novelty of this work. I wonder if the caption could be shortened. It is currently pretty long.
* * *

---

## Referee Comment (RC3) · Anonymous Referee #3 · 19 Jul 2017

This manuscript uses ISI-MIP streamflow simulations to explore the joint future of hydrological extremes (low and high flows). This is a quite relevant topic for HESS, but I have concerns about the novelty of the study, given the wealth of already published papers on hydrological extremes derived from ISI-MIP simulations. Furthermore, the statistical analysis is not (yet) convincing in my view – and choices are not justified (enough) – to bring the paper to a level where it could be published in HESS. All comments below have been initially drawn before I read comments from the two other referees, and I then added references to these in order to highlight common assessments or suggestions.

**Major Comments**

1. As mentioned above, and as already noted by Referee 1, there is little novelty in the topic and dataset used compared to previously published literature (especially to Giuntoli et al., 2015), and the little amount of novelty is not pushed forward in the manuscript. In my view, there are two new contributions: (1) the comparison between two contrasted RCPs, and (2) the quantification of absolute changes in high/low flow indices and their joint analysis. I agree with Referee 1 proposal to better highlight the manuscript's contributions, but I fear there are other issues that need to be tackled first.

2. The quantification of changes is, as already noted by the two other referees, first quite questionable in terms of wordings: high and low flow indices simply cannot be identified to flood and drought indices. Changes throughout the manuscript (including title) are therefore required. Furthermore, the authors consistently use the wording of streamflow in the manuscript, but I believe that the variable used is the (unrouted) runoff, as in previous related works on ISI-MIP data (Prudhomme et al, 2014; Giuntoli et al., 2015), and on the contrary to other works on (large) river basins (see e.g. Pechlivanidis et al., 2017; Vetter et al., 2017). This has serious implications for interpreting results in terms of floods and droughts (see the recent work by Zhao et al., 2017).

3. The normalization procedure is probably interesting for positive variables like streamflow, as it makes multiplicative factors symmetrical with respect to zero. Multiplying (resp. dividing) present-day values by 3 results in a value of 1/2 (resp. -1/2). However, the lack of experience with dealing with such an index makes it rather difficult to interpret values. The way values converge towards 1 or -1 is for example not intuitive. The reader should at least be accompanied through this kind of basic examples.

[Figure]

4. The joint analysis of changes in low flow and high flow indices is potentially attractive. However, I don't understand why the analysis is restricted to quadrants (cf. Figure 5) when all data are available for continuous assessments over the two indices (see Teuling et al., 2011a, b). This is in my view an oversimplification of the problem. You cannot identify with the quadrants a region with a small drought increase and a large flood increase (whatever that means). Moreover, the multi-model average is, as pointed out by Referee 1, potentially quite misleading. This is all the more problematic that there is a confusion (at least of the reader) when dealing with statistical significance. At several places in the manuscript, one may expect some tests for example on the sign of change within the multimodel ensemble (see the latest IPCC report), and not (only) the significance of changes between 30-year averages of future and present period for single models. Many detailed and interesting statistical analyses could be performed with this dataset by applying ANOVA techniques, and by for example deriving individual maps of GCMs/GHMs effects (in the ANOVA sense) on joint changes in low/high flow indices. This would avoid using latitude-averaged plots that do not convey in my view the most relevant information. For example, it is not possible on Figures 2, 3, and 4 to compare the spatial variance (along any given latitude) from the variance among GCMs/GHMs/combination of GCMs and GHMs (depending on the figure).

5. This also leads to my last major comment. I don't really understand why this study is restricted to only 5 GHMs. Statistical techniques are indeed available to take account of different sample sizes in ANOVA contexts (see for example Giuntoli et al., 2015). Furthermore, there is no justification in the manuscript on the choice of these specific 5 GHMs, and this has already been pointed out by Referee 1. This thus appears as a subjective and therefore negative choice for building confidence in results from this "ensemble of opportunity".

[Figure]

**Specific comments**

1. P1L16: percentage with respect to what period?

2. P2L14: Please make explicit what you mean by "impact". The hierarchy of impacts (for example in terms of monetary loss) is indeed highly dependent on the anthropogenic system under study.

3. P2L19: I believe this is about "average runoff". Please specify.

4. P5L21: The normalization is announced and summarized here whereas it is described only much later on (P6 L3 ff.). Please reorganize the paragraphs.

5. P6L12-14: This is hardly understandable. Please consider giving the actual equations.

6. P7L15-19: These figures are redundant with Table 1. Please rephrase.

7. P8L5, "with high agreement": Could you explain what you mean exactly here?

8. P8L19, "fluctuations": Again, what do you mean here? Fluctuations in time, latitude, other? Please specify.

9. P8L23, "mean": over space, latitude? Please be more specific.

10. P9L8, "statistically different": What is the test used here? Please be more specific on your statements.

11. P10L11, "statistically significant": see above.

12. P11L16-17: This final sentence is rather ambiguous and wrongly suggests a 200

**Technical corrections**

1. P2L3, "to be intensified": please rephrase

2. P2L8, "dictation": please rephrase

3. P4L11: "increase"

4. P6L17, "remained": please rephrase

5. P8L14, missing "in" after "flux"

**References**

Giuntoli, I., Vidal, J.-P., Prudhomme, C. Hannah, D. M.: Future hydrological extremes: the uncertainty from multiple global climate and global hydrological models, Earth System Dynamics, 6(1), 267-285, doi: 10.5194/esd-6-267-2015, 2015

Pechlivanidis, I. G., Arheimer, B., Donnelly, C., Hundecha, Y., Huang, S., Aich, V., Samaniego, L., Eisner, S. Shi, P.: Analysis of hydrological extremes at different hydroclimatic regimes under present and future conditions, Climatic Change, 141(3), 467-481, doi: 10.1007/s10584-016-1723-0, 2017

Prudhomme, C., Giuntoli, I., Robinson, E. L., Clark, D. B., Arnell, N. W., Dankers, R., Fekete, B. M., Franssen, W., Gerten, D., Gosling, S. N., Hagemann, S., Hannah, D. M., Kim, H., Masaki, Y., Satoh, Y., Stacke, T., Wada, Y. Wisser, D.: Hydrological droughts in the 21st century, hotspots and uncertainties from a global multimodel ensemble experiment, Proceedings of the National Academy of Sciences, 111(9), 3262-3267, doi: 10.1073/pnas.1222473110, 2014

[Figure]

Teuling, A. J., Stöckli, R. Seneviratne, S. I.: Bivariate colour maps for visualizing climate data. International Journal of Climatology, 31(9), 1408–1412, doi: 10.1002/joc.2153, 2011a

Teuling, A. J.: Technical note: Towards a continuous classification of climate using bivariate colour mapping, Hydrology and Earth System Sciences, 15(10), 3071-3075, doi: 10.5194/hess-15-3071-2011, 2011b

Vetter, T., Reinhardt, J., Flörke, M., van Griensven, A., Hattermann, F., Huang, S., Koch, H., Pechlivanidis, I. G., Plötner, S., Seidou, O., Su, B., Vervoort, R. W. Krysanova, V.: Evaluation of sources of uncertainty in projected hydrological changes under climate change in 12 large-scale river basins, Climatic Change, 141(3), 419-433, doi: 10.1007/s10584-016-1794-y, 2017

Zhao, F., Veldkamp, T. I. E., Frieler, K., Schewe, J., Ostberg, S., Willner, S., Schauberger, B., Gosling, S. N., Müller Schmied, H., Portmann, F. T., Leng, G., Huang, M., Liu, X., Tang, Q., Hanasaki, N., Biemans, H., Gerten, D., Satoh, Y., Pokhrel, Y., Stacke, T., Ciais, P., Chang, J., Ducharne, A., Guimberteau, M., Wada, Y., Kim, H. Yamazaki: The critical role of the routing scheme in simulating peak river discharge in global hydrological models, Environmental Research Letters, 12, 075003, doi: 10.1088/1748-9326/aa7250, 2017

---

## Author Response (AR1)

Dear reviewers,

Thank you very much for your valuable comments. We have applied the suggested revisions to the manuscript and have responded to comments, point by point, here.

General Remarks (some parts may be repeated in point-by-point responses to the comments)

Regarding the definitions of flood and drought, we agree that there are many different indices in the literature that can be used, and which may be well suited for different purposes. However, the abundant of indices also adds to the complexity of selection of appropriate indices, and we need to select one so that an analysis at this scale is practical. In case of our selection, we may note that United States Geological Survey (USGS) in its annual streamflow reports uses the 95th and 5th percentiles of streamflow as thresholds for high and low flow studies (Jian et al., 2015). Other studies also have used 5th and 95th percentiles to define streamflow extremes (Koirala et al., 2014). Earlier studies have used the 95th percentile of streamflow as a threshold for flood definition in gridded streamflow data (Wu et al., 2012, 2014). The 5th percentile of streamflow has also been used as a threshold for drought definition (Ellis et al., 2010; Sprague, 2005). Such a definition for flood threshold is being used in daily flood monitoring maps produced by the USGS (https://water.usgs.gov/floods/) (please refer to the map's legend: 95th-98th percentile and 99th percentile categorization) and in the NASA-funded gridded 1/8th degree resolution Global Flood Monitoring System (GFMS) (http://flood.umd.edu/) (please refer to General Description). We have added these references to percentile-based streamflow thresholds to the revised manuscript.

However, in response to the comments made by the reviewers, we have omitted the words flood and drought from the revised title and use "streamflow extremes" instead. Accordingly, we study changes in 95th and 5th percentiles of streamflow and only refer to changes in those extremes as increase in potential for flood and drought hazards. We edited the manuscript accordingly.

We see in the comments that the referees show interest in the joint analysis of high and low flow extremes, but also recommend to better describe the novelty of this study in the manuscript. Accordingly, we have added paragraphs to the introduction to emphasize on the knowledge gap that we are aiming to fill. To summarize a few, Giuntoli et al. 2015 studies changes in *frequency* of *un-routed* runoff (dimension  $[L.T^{-1}]$ ) extremes under *one* warming scenario, while we study changes in *magnitude/intensity* of *routed* runoff (streamflow, dimension  $[L^3.T^{-1}]$ ) extremes under *two* different warming scenarios. Un-routed runoff may not be directly used for flooding study, but routed runoff (streamflow) has been used for this purpose in the literature instead (for instance: Koirala et al., 2014 and Hirabayashi et al. 2013). So although Giuntoli et al. 2015 is closest to ours in some respects, there are critical differences. Prudhomme et al 2014 also studies un-routed runoff, not streamflow. Koirala et al. 2014 studies 5th and 95th percentiles of streamflow, but routed by only one state-of-the-art GHM, and hence, does not account for GHM uncertainties. Hirabayashi et al. 2013 also uses a single state-of-the-art GHM, to study the *frequency* of extremes.

We study the change in the *magnitude/intensity* of the high and low extremes to understand the projected changes in the distribution of streamflow over the next century, while earlier studies (Giuntoli et al. 2015 for instance, and the others referred in the comments by reviewers) study changes in the *frequency* of streamflow passing the high and low extreme threshold. However, the thresholds used in those studies are defined based on the historical (20th century) distribution of streamflow. We find that the distribution of the streamflow is projected to change in the 21st century, and hence, the 95th/10th/5th percentiles used by these studies will not hold over the next century. Therefore, studying the changes in the *magnitude* of streamflow extremes can provide better interpretability of projected changes in the streamflow distribution compared to the *frequency*. Also, in the current work we study changes in flood and drought risks (both high and low flow extremes) together to understand the changes in streamflow distribution and simultaneous vulnerability profiles to different types of hydrological risk in different regions.

We also study the changes under two emission scenarios (RCP2.6 and RCP8.5) to compare the impact of different levels of warming on streamflow distribution and consequently flood and drought risks (Giuntoli et al. 2015 uses only one (RCP8.5) warming scenario). We also calculate the number people affected by changes in different regions to understand whether the areas of increased risk coincide with highly populated regions.

Thank you.

**Author Response References:**

Ellis, A. W., Goodrich, G. B. and Garfin, G. M.: A hydroclimatic index for examining patterns of drought in the Colorado River Basin, Int. J. Climatol., 30(2), 236–255, doi:10.1002/joc.1882, 2010.

Giuntoli, I., Vidal, J. P., Prudhomme, C. and Hannah, D. M.: Future hydrological extremes: The uncertainty from multiple global climate and global hydrological models, Earth Syst. Dyn., 6(1), 267–285, doi:10.5194/esd-6-267-2015, 2015.

Jian, X., Wolock, D. M., Lins, H. F. and Brady, S.: Streamflow of 2015—Water year national summary, U.S. Geological Survey Fact Sheet 2016–3055, 6 p., doi:10.3133/fs20163055. [online] Available from: https://waterwatch.usgs.gov/2015summary/, 2015.

Koirala, S., Hirabayashi, Y., Mahendran, R. and Kanae, S.: Global assessment of agreement among streamflow projections using CMIP5 model outputs, Environ. Res. Lett., 9(6), 64017, doi:10.1088/1748-9326/9/6/064017, 2014.

Hirabayashi, Y., Mahendran, R., Koirala, S., Konoshima, L., Yamazaki, D., Watanabe, S., Kim, H. and Kanae, S.: Global flood risk under climate change, Nat. Clim. Chang., 3(9), 816–821, doi:10.1038/nclimate1911, 2013.

Sprague, L. A.: Drought Effects on Water Quality in the South Platte River Basin, Colorado, J. Am. Water Resour. Assoc., 41(1), 11–24, doi:10.1111/j.1752-1688.2005.tb03713.x, 2005.

Wu, H., Adler, R. F., Hong, Y., Tian, Y. and Policelli, F.: Evaluation of Global Flood Detection Using Satellite-

Based Rainfall and a Hydrologic Model, J. Hydrometeorol., 13(4), 1268–1284, doi:10.1175/JHM-D-11-087.1, 2012.

Wu, H., Adler, R. F., Tian, Y., Huffman, G. J., Li, H. and Wang, J.: Real-time global flood estimation using satellite-based precipitation and a coupled land surface and routing model, Water Resour. Res., 50(3), 2693–2717, doi:10.1002/2013WR014710, 2014.

**Anonymous Referee #1**

Received and published: 27 June 2017

Review of "Global change in flood and drought intensities under climate change in the 21st century".

This paper aims at understanding how climate change will manifest in changes in flood and drought conditions globally. To pursue this, the authors collected ISI-MIP streamflow projections (which are based on bias corrected output of five global climate models (CMIP5), and five hydrological models) at a 0.5x0.5 spatial scale. Climate scenario's that are explored include RCP8.5 (which assumes emissions continue to rise throughout the 21st century) and RCP2.6 (which assumes emissions peak between 2010-2020, with emissions declining substantially thereafter).

To quantify changes in flood and drought conditions, the authors calculated the "normalized change" which they defined as the difference between mean percentiles (5th for low flows and 95th or high flows) of 20th century flow conditions compared to 21st century flow (normalized by the sum of 20th and 21st flow percentiles). Calculations are performed for areas where streamflow >0.01 mm/d, and exclude Greenland. Global maps of changes are provided.

The results suggest that: (i) Globally, both floods and drought are expected to intensify. (ii) in some regions (especially several highly-populated areas), both flood and drought intensity will intensify (iii) Especially, in northern high latitudes flood intensities will increase. (iv) RCP8.5 leads to nearly 2 times higher impact on changes in flood and drought intensities compared to RCP2.6. (v) Hydrological models have a larger contribution to uncertainty in the projections, compared to the effect of different climate models.

General comments

Understanding the nature of future floods and drought, and the effect of changing climate conditions on these hydrological extremes is very relevant for HESS. The analysis is rather straightforward and reasonably well-explained.

However, I do have several (major) concerns that need to be addressed before I recommend publication of this manuscript in HESS:

1) What makes the metric you define to characterize changes in flood and low intensities appropriate for the problem that you address? While the 5th and 95th flow percentiles are surely representing higher and lower flow conditions, they are not real hydrologic extremes. For example, if the 95th percentile represents flood conditions, every grid cell would experience \_18 flood days per year. Would it not be much more useful to quantify changes in the more extreme conditions (e.g. 99th percentile?). It seems that this choice arises from Giuntoli el al (2015)

(except that they use the 10th percentile for low flow), but does that warrant that it actually is an appropriate metric to use?

We agree that there are many different indices in the literature for flood and drought definitions that can be used. However, the abundant of indices also adds to the complexity of selection of appropriate indices, and we need to select only one of those so that an analysis in this scale is practical. In case of our selection, we may note that United States Geological Survey (USGS) in its annual streamflow reports uses the 95th and 5th percentiles of streamflow as thresholds for high and low flow studies (Jian et al., 2015). Other studies also have used 5th and 95th percentiles to define streamflow extremes (Koirala et al., 2014). Earlier studies have used the 95th percentile of streamflow as a threshold for flood definition in gridded streamflow data (Wu et al., 2012, 2014). The 5th percentile of streamflow has also been used as a threshold for drought definition (Ellis et al., 2010; Sprague, 2005). Such a definition for flood threshold is being used in daily flood monitoring maps produced bv the USGS (https://water.usgs.gov/floods/) (please refer to the map's legend: 95th-98th percentile and 99th percentile categorization) and in the NASA-funded gridded 1/8th degree resolution Global Flood Monitoring System (GFMS) (http://flood.umd.edu/) (please refer to General Description). We have added these references to percentile-based streamflow thresholds to the revised manuscript.

2) A lot of literature is already available on the topic of projected flood and drought (/low flow) changes. While you cite several papers in the introduction, it remains unclear to me which knowledge gap your paper fills compared to earlier work. I made a list of detailed suggestions in the comments below, which may help to address this issue.

However, right now when I read the abstract I do not see understand knowledge gap you address, when I read the introduction no I get some idea of what has been done before but still no clear niche is identified that you fill. Any discussion is a description of the results rather than how we better understand global flood and drought changes, and in the conclusions, you summarize findings, but still I have difficulty how this relates to the vast amount of literature on previous flood changes (also what's new compared to Arnell et al. 2016, Alfieri et al. 2015, 2017, Giuntoli el al. 2015?). While I do not doubt that are new things in your paper, they need to be clearly identified.

We have added more paragraphs to the introduction to emphasize on the knowledge gap that we are aiming to fill. To summarize a few, Giuntoli et al. 2015 studies changes in *frequency* of *un-routed* runoff (dimension  $[L.T^{-1}]$ ) extremes under *one* warming scenario, while we study changes in *magnitude/intensity* of *routed* runoff (streamflow, dimension  $[L^3.T^{-1}]$ ) extremes under *two* different warming scenarios. Un-routed runoff may not be directly used for flooding study, but the routed runoff (streamflow) has been used for this purpose in the literature instead (for instance: Koirala et al. 2014 and Hirabayashi et al. 2013). So although Giuntoli et al. 2015 is closest to ours in some respects, there are critical differences. Prudhomme et al 2014 also studies

un-routed runoff, not streamflow. Koirala et al. 2014 studies 5th and 95th percentiles of streamflow, but routed by only one state-of-the-art GHM, and hence, does not account for GHM uncertainties. Hirabayashi et al. 2013 also uses a single state-of-the-art GHM, to study the frequency of extremes. We study the change in the magnitude/intensity of the high and low extremes to understand the projected changes in the distribution of streamflow over the next century, while earlier studies (Giuntoli et al. 2015 for instance, and the others referred in the comments by reviewers) study changes in the *frequency* of streamflow passing the high and low extreme threshold. However, the thresholds used in those studies are defined based on the historical (20th century) distribution of streamflow. We find that the distribution of the streamflow is projected to change in the 21st century, and hence, the 95th/10th/5th percentiles used by these studies will not hold over the next century. Therefore, studying the changes in the magnitude of streamflow extremes can provide better interpretability of projected changes in the streamflow distribution compared to the *frequency*. Also, in the current work we study changes in flood and drought risks (both high and low flow extremes) together to understand the changes in streamflow distribution and simultaneous vulnerability profiles to different types of hydrological risk in different regions. We also study the changes under two emission scenarios (RCP2.6 and RCP8.5) to compare the impact of different levels of warming on streamflow distribution and consequently flood and drought risks (Giuntoli et al. 2015 uses only one (RCP8.5) warming scenario). We also calculate the number people affected by changes in different regions to understand whether the areas of increased risk coincide with highly populated regions.

3) Related to the previous points: the rationale behind some choices in the analysis is missing. Can you answer: (i) why we would be interested in flood and drought changes simultaneously.

Flood and drought risks are studied using the high and low extremes. Studying the change in magnitude of the high and low extremes together shows the change in the distribution of the streamflow.

(ii) why the metric you choose are appropriate to characterize what you do

We explained our choice of indices earlier here.

(iii) why correlating population density and change in flood hazard is meaningful.

We are interested to know whether the areas of increased risk coincide with highly populated regions (and hence, how many people are at risk).

(iv) why looking at "intensity" is novel and important compared to "frequency (which is already available using a similar approach).

We study the change in the *magnitude* of the high and low extremes to understand the projected changes in the distribution of streamflow over the next century, while earlier studies

(Giuntoli et al. 2015 for instance, and the others referred in the comments by reviewers) study changes in the *frequency* of streamflow passing the high and low extreme threshold. However, the thresholds used in those studies are defined based on the historical (20th century) distribution of streamflow. We find that the distribution of the streamflow is projected to change in the 21st century, and hence, the 95th/10th/5th percentiles used by these studies will not hold over the next century. We argue that studying the changes in the magnitude of streamflow extremes provides us with better understanding of projected changes in the streamflow distribution compared to the frequency.

(v) why do we also need to look at changes in median flow (when the purpose seems extreme flows) I do not try to imply that these choices are not well thought out and relevant. However, I do think you need to take the reader by the hand in why such choices are made

One goal of our work is to also study changes in the distribution of streamflow time series. Studying mean flow along with high and low extremes enables us achieve that goal better. We have reflected this in the revision.

4) 0.5 x 0.5 degrees and daily forcing seems like very large spatial and temporal scales to resolve the hydrology of flood processes. I understand it is "the best you can work with" right now if you want to understand flood changes globally for the entire 21st century.

However, can you better reflect on how this actually affect the degree to which you can resolve flood changes? Many different flood generating processes cannot be represented at these scales. For example, flash floods (e.g. I expect you need subdaily P for this?) or snowmelt driven floods (e.g. I expect that you need to parametrize the sub-grid heterogeneity of snow conditions) seem challenging, while in many places these processes are very important (e.g. see Berghuijs et al., 2016). Since there essentially is very little science in the paper (basically there are no rejectable hypotheses you test, and the paper provides a summary of available data), I think you need to say a few useful things on this topic, such that it still meets the standards of a journal like HESS.

We actually do not investigate hydrology of flood processes, but change in streamflow extremes under climate change that results in change in flood risks. As you said, it may not be feasible to study flood process in such coarse resolution, and in global scale. For that purpose, one may resort to Reginal Climate Models and limit the study to basin scale. It may not be feasible to study flash floods in global scale, since these analyses depend on the local land cover and soil type (and in shorter time scales), which are not accurately modeled in Global Climate Models. This is one of the reasons that we use multiple GHMs that use different routing schemes for streamflow generation to at least address those types of uncertainties to some degree. However, by understanding the projected changes in streamflow extremes in large scale over a region, one may use this knowledge to better calibrate the regional models to study flash floods in finer spatial and temporal scale. Our study is more similar to that of Koirala et al. 2014 and/or Hirabayashi et al. 2013. As we stated earlier, we added paragraphs to the Introduction section of

the manuscript to explain better why we are interested in this type of analysis and what knowledge gap we are aiming to fill.

5) I suggest to be careful with the using low flows and drought interchangeably. Low flows and droughts are not the same. You quantify changes in low flow conditions, not in drought conditions (which reflect some deviation compared to the normal flow conditions of a catchment during a particular time of the year). These two different concepts should not be mixed. (e.g. see Van Loon, 2015). To some degree the same applies for high flows and floods. However, this difference seems less important, since those two concepts are more closely related.

We have changed the terminology in the revision, and link the change in low flow to drought *risk*, instead of direct change in drought *intensity*.

6) The changes in flood and drought indices per latitude intrigue me (Fig 2). How can they virtually be the inverse of another (when organized per latitude)? Is this the physical reality or an artifact of how this study quantifies changes in these aspects?

Surely, floods and drought within one area can be connected (since they are part of the same climate and landscape system). However, (in my opinion) this almost oneon-one inverse pattern seems to require some attention.

This is actually expected. We define the increased flood indicator as increase in P95 and increased drought indicator as decrease in P5, so by increase in all percentiles of flow the flood and drought indicators should show the opposite direction of change (same for decrease in all percentiles of flow). If all percentiles of flow increase in a grid cell, it means that the flood indicator would increase (P95 increase) but drought indicator decrease (P5 increase). However, there are regions that the flood and drought indicators both increase or both decrease or have different rate of change, due to change in the distribution of the streamflow; such grid cells are located in this study.

7) You quantify the percentage of land area that undergoes significant streamflow change. However, the two-sample t-test that you use for this (as explained in the supplementary material) seems inappropriately used. Why would scaling this relationship by the streamflow time series variances be appropriate here? Should they not be scaled by some measure of (e.g. between year) variability in the hydrologic extremes flow?

This is actually what we have done. We do the calculations directly on the extremes times series, not the mean time series. We edited the Supplementary Materials' text to clarify this.

8) By presenting only mean values of results (e.g. the mean magnitude of flood and drought changes) the reader has no idea from what distribution of changes these mean values are derived. This seems rather basic information that can easily be presented in will provide valuable insight for the reader. Right now, some of the numbers you use in key points of your

paper are difficult to interpret because we have no idea if they represent many small changes and a few extreme ones, or because they represent consist medium size changes.

We report mean value of the results in the abstract to summarize the global findings, and in the text to discuss the changes in a broader scale. However, we do provide the global maps of changes in each indicator, which show the magnitude of changes (represented by color saturation) in each grid cell and under each warming scenario. The scatter plots in Figure 2c&d also show the distribution of small and large changes under each scenario. We also show the results averaged by latitude to show the averaged changes over different climate boundaries (high latitudes, mid latitudes, tropics and subtropics, etc). We also provide the global maps for each of the models individually in the Supplementary Materials.

9) Please address the list of comments I provide below. (Several of these comments go beyond small technical details.)

detailled comments

Page 1 Line 11-12. Consistent with my main comments above, it would be very useful to have a transition sentence that actually introduces the knowledge gap in flood and drought risk projections that the paper aims to fill. Preferable reflect on that goal in the end of the abstract.

As we stated earlier, we added paragraphs to the Introduction section and revised the manuscript to explain better what knowledge gap we are aiming to fill.

Line 16: without a definition of what aspects of these hydrological extremes you actually look at (e.g. duration, or magnitude, or both)) these precise percentages not very useful. Being more specific in line 12 may resolve this issue. The same problem applies to all other percentages provided in the abstract. (or the statement in lines 22-24)

As answered in other comments, we have revised the manuscript and the terminology to clarify our findings.

Line 17-19: "the averaged rates of increase" (can) suggest that you exclude places where it reduced? Or is this the average of all increases and decreases globally?

Yes, this is the average of change in those areas that experience increase. Average rates of increase and decrease are reported separately, as also shown in Table 1.

Line 19: "potential risk" or "are projected" (since I guess all areas are under the "potential risk"?)

We edited it to "projected risk".

Line 20: "rate" or "change" (or "increase")?

We edited to "average rates of changes".

*Line 21. Semi-column or just start a new sentence?*

Since it is referring to the regions experiencing simultaneous increase in flood and drought, as reported in the previous sentence, a semi-colon is used.

*Line 26-29: It seems odd to me that the paper suddenly talks about changes in streamflow (I guess that means mean runoff?) while the rest of the paper is about extremes?*

We are talking about streamflow, not runoff (one may define streamflow as routed runoff though). We use streamflow here in general, since both low and high extremes as well as mean flow (whole streamflow spectrum) show the reported changes in the reported areas.

**Page 2**

Line 15-18: Sure: more extreme P can lead to more extreme runoff. However, there is so much more going on that dictates runoff response (e.g. antecedent moisture conditions in floods etc). Would it be worth to say one or two things about other mechanisms that underlie floods? In many places, there is a disparity between extreme rainfall and flooding, or between lowest P and lowest Q, since so many other factors are also important (e.g. seasonal moisture conditions). Emphasizing which other processes are important may help to understand the reader what the added value is of adding the GHM's to the game (since they at least theoretically should represent all these processes that go beyond extreme P). Nor can I logically connect more extreme high P to more extreme drought (without some extra information about changes in dry spells, or hydrologic partitioning)

We do not specifically describe how more extreme P results in more extreme runoff because it this is not the main focus of our study and we do not want to lengthen our paper unnecessarily. Our focus is mainly on how the streamflow distribution and extremes change. We provide references for these statements to support the claim, and refer the readers to appropriate papers should they be interested in reading more about this topic. However, we do describe the impact of GHMs in the flow outing process and their implications for uncertainties in streamflow simulation, in the Introduction section.

Line 19-29: I do not see why the paper needs to talk about changes in "mean streamflow conditions" since it distracts from what you're really interested in (which are the hydrologic extremes).

Since in majority of areas, high and low extremes and mean flow show similar directions of change, an introduction in general change in streamflow as projected by GCMs would be useful. However, some regions show opposite direction of change in high and low extremes, which can translate as increase in both flood and drought risks. Simultaneous study of high and low extremes, as done in our study, helps to locate such regions.

*Line 31-33: Sure, that a decrease in P can decrease runoff. However (like you give with the following example) you can also think of conditions where this does not apply.*

This is exactly our point: to show that mean runoff may not be an appropriate indicator of change in water stress (and hence, drought), but changes in distribution of runoff should be studied.

**Page 3**

Line 1-5: Ok, I understand that there may not be many studies that use ensembles. However, still that does not answer the question of what knowledge gap you can fill with your approach. What do we not understand because we haven't run particular ensemble projections yet?)

As we stated earlier, we added paragraphs to the Introduction section and revised the manuscript to explain better what knowledge gap we are aiming to fill.

Line 4-5: The detection of areas that are expected to experience both more floods and drought sounds interesting at first, but what is again the knowledge gap that the paper fills compared to earlier work, and what is the merit in identifying these at the same time (there are reasons why this can be valuable, but they need to be presented to the reader).

Studying floods and droughts (change in magnitude of high and low extremes) together enables us to better understand changes in distribution of streamflow time series in different regions, which was not achievable through studying only one tail of the extremes. Also, as we explained earlier here, the previous studies have investigated changes in frequency of extremes, which as we argued earlier, change in distribution is better achieved through studying changes in the magnitude/intensity. We have explained this more in detail in the revision.

Lines 5-17: put these findings into context of the novel thing you're going to expose/test. Right now, it reads like a random list of previously reported streamflow changes, which are unclear why they're directly relevant to the paper.

We have edited the text in the revision.

*Lines 18-20: Maybe a reference (or two) can help to support this statement?*

References added.

*Line 20: remove "trend" (since there may not be one)*

Done.

*Line 13-14: Be very explicit to the reader what the difference between "frequency" and "intensity" is, and emphasize why this difference is relevant.*

Earlier studies investigate changes in frequency, instead of magnitude/intensity. Thresholds used in those studies for extremes are defined based on the historical ( $20^{th}$  century) distribution of streamflow. We find that the distribution of the streamflow is projected to change in the  $21^{st}$  century, and hence, the  $95^{th}/10^{th}/5^{th}$  percentiles used by these studies will not hold over the next

century. We argue that studying the changes in the *magnitude* of streamflow extremes provides us with better understanding of projected changes in the streamflow distribution compared to the *frequency*. We have explained this more in detail in the revision.

Line 13: "this study" may be unclear because it can refer to your own work or the work of Giuntoli

We edited it to "the aforementioned study".

Line 15: It may be worth to start state "Here we" and then list the "goal", rather than directly go into the "methods". This will make the list of subsequent steps outlined in the rest of the introduction much more logical. For example, right now it sounds fun that you also investigate the link with human populations, but I have no idea (or at least it's up to my own guess!) why you'll be doing this.

We rearranged the paragraph.

Page 5 Line 13: why are these five GHM's selected? (after line 15-17 that question still stands)

We had initially selected five GHMs to have equal number of GCMs and GHMs so they contribute equally to the uncertainties. However, we removed this justification from the manuscript in response to the reviewers' comments. These GHMs have been selected by earlier studies as well, for flood change investigations.

Line 18-19: do you have any references that show this, or did this only appear in your own work?

This is problem that we faced during our calculations (and mathematics dictates that other similar studies may face this issue as well, since the GCMs show very large changes in northern latitudes). We used the normalized change to solve the issue.

Lines 19-21: consider rewriting this sentence

The sentence is rewritten for better clarity and flow.

Lines 18-27: it seems a bit confusing to justify normalization before you define the metric that you adopted.

We moved the paragraph to after definition of the metric.

*Line* 28 – *Line* 2: *Is* "no change" also an option?

The four categories only show the grid cells with increase or decrease in the indicators.

*Line 3-14: Why did you choose the 5th and 95th percentile and not "real extremes" (see earlier comment above)*

Answered earlier.

Line 16: 200?

2000 (edited).

Lines 26-29: And what about any places where there are insignificant changes?

Percentage of grid cells with statistically significant change at 95% confidence level is reported in the results section.

Lines 30-3: why don't you use the absolute value (and then you don't need to separate by quadrant).

As explained in the Methods section, using absolute values would result in a large positive global trend (due to the large relative changes in high latitudes) indicating a significant increase in flood in global average, which is not realistic according to normalized averages.

Line 3-7: Why would you even bother to try that method? It seems like this method is just less logical at the start (because it is very sensitive to absolute runoff changes between models), and hence should not be considered at all?

But as we stated there, that method also yields very similar results, proving that the final results are not sensitive to the order of averaging.

Lines 8-13: Your results suggest that 95% of the projected flood changes are significant, but what does that really mean? Does that imply that for 95% of the grid cells you are very certain about the projections? Or does it mean that model projections may show a significant change, but all other biases and uncertainties not accounted for may lead to much lower certainties of projected change?

Statistical significance at 95% confidence level does not actually mean that 95% of the changes are significant. We calculate the statistical significance of change for all of the grid cells, and if the statistical significance of change in a grid cell is over 95% we say "we are very certain about the projections". Some cells might show 80% or 10% of confidence. We have reported the percentage of grid cells that show over 95% confidence level in the projected change.

Also (in the supplementary material), why would "streamflow time series variances" be a relevant scale of variance here (rather than something like the variance of annual maxima or Q95).

The variances are actually calculated for the P5 and P95 time series (over the 20th century and 21st century). We edited the Supplementary Materials to reflect this clarification.

Line 16-18: you need to show the distribution of changes, rather than just the mean value. Right now I have no idea if the mean results from consist small changes, or a few very big changes.

The distribution of changes is shown in Figures 2 and 5.

Line 19: why bother with median flows? I thought this paper was about the extremes?

One goal of our work is to also study changes in the distribution of streamflow time series. Studying mean flow along with high and low extremes enables us achieve that goal better.

**Page 8**

Figure 2: The changes in flood and drought indices per latitude intrigue me. How can they virtually be the inverse of another (when organized per latitude)? Is this a physical reality or an artifact of how this study quantifies changes in these aspects?

This is actually expected. We define the increased flood indicator as increase in P95 and increased drought indicator as decrease in P5, so by increase in all percentiles of flow the flood and drought indicators should show the opposite direction of change (same for decrease in all percentiles of flow). If all percentiles of flow increase in a grid cell, it means that the flood indicator would increase (P95 increase) but drought indicator decrease (P5 increase). However, there are regions that the flood and drought indicators both increase or both decrease or have different rate of change, due to change in the distribution of the streamflow; such grid cells are located in this study.

Figure 5: it is impossible to read the scale bar in the far bottom left (on a printed page).

We enlarged the text in the revision.

**Page 10**

Line 30: you were not interested in decreases?

Such regions are not under threat (they are shown in the maps though).

Line 31 because people live in a grid cells where floods increase does not mean they are affected. That depends on many other factors. Correct?

We can say they are affected, but it is correct that the effect may not be serious for some regions.

**Page 11**

While I appreciate, you repeat all the main results of the paper, I think the paper really needs to reflect on what we learned compared to earlier work, rather than list what came out of some modeling exercises.

We revised the Conclusion Section to reflect our findings better.

Table 1: without information on the distribution of changes, I have no idea about what these mean values of change represent.

The scatter plots in Figure 2c&d also show the distribution of small and large changes under each scenario. We also provide the global maps of changes in each indicator, which show the magnitude of changes (represented by color saturation) in each grid cell and under each warming scenario.

Reference list: what does : "(80-.)" do in several references?

It was an error caused by the references management software. Fixed.

**Anonymous Referee #2**

Received and published: 11 July 2017

The article by Asadieh and Krakauer investigates the very topical issue of flood and drought changes under future climate conditions. The topic has been subject to a large number of studies in the past few years, many of them based on the same set of GCM-GHM combinations from the ISI-MIP initiative, so it is difficult to find some unexplored topic of research in this area. However, this work is based on an interesting idea of comparing together increases in droughts and flood intensity and frequency under future climate, and I think it has potential for being

published. The writing style is up to international standards and the article is compact, hence I don't see room for shortening.

My main concern is the misleading use of the terms "floods" and "droughts" throughout the article, for indicating high and low streamflow quantiles which are not really extremes, and certainly not linked to actual flood or drought events. Floods are normally linked to much higher quantiles, and in addition, they depend on the local vulnerability.

Streamflow droughts (which by the way should be specified in the article, as meteorological and agricultural droughts are calculated differently) are also not as simple as a connection to the streamflow quantile, but they depend on the duration and intensity of the droughts. My suggestion is to clarify well through the article (e.g., p4 118-21, p5 122-25, p6, and in general in the results) and in the title that the aim is to "high and low streamflows" rather than floods and droughts. Interestingly, only in the caption of Fig 1 did the authors write a warning about linking those streamflow quantiles to actual floods and droughts.

We agree that there are many different indices in the literature for flood and drought definitions that can be used. However, the abundant of indices also adds to the complexity of selection of appropriate indices, and we need to select only one of those so that an analysis in this scale is practical. In case of our selection, we may note that United States Geological Survey (USGS) in its annual streamflow reports uses the 95th and 5th percentiles of streamflow as thresholds for high and low flow studies (Jian et al., 2015). Other studies also have used 5th and 95th percentiles to define streamflow extremes (Koirala et al., 2014). Earlier studies have used the 95th percentile of streamflow as a threshold for flood definition in gridded streamflow data (Wu et al., 2012, 2014). The 5th percentile of streamflow has also been used as a threshold for drought definition (Ellis et al., 2010; Sprague, 2005). Such a definition for flood threshold is used in daily flood monitoring maps produced by the USGS being (https://water.usgs.gov/floods/) (please refer to the map's legend: 95th-98th percentile and 99th percentile categorization) and in the NASA-funded gridded 1/8th degree resolution Global Flood Monitoring System (GFMS) (http://flood.umd.edu/) (please refer to General Description). We have added these references to percentile-based streamflow thresholds to the revised manuscript.

However, in response to the comments made by the reviewers, we have omitted the words flood and drought from the revised title and use "streamflow extremes" instead. Accordingly, we refer to change in 95th and 5th percentiles of streamflow as change in flood and drought "*risk*", instead of direct change in flood and drought "*intensity*". We edited the manuscript accordingly.

**Specific comments**

P1 111-12: This sentence reads more like a finding rather than an introduction. I'd move it to the introduction and support it with some references.

This statement is reported in the Introduction section and corresponding references are given to support it. We have included this sentence in the abstract as well to familiarize the readers with the general topic that we are investigating.

*P2 l17-18: I suggest complementing the list with the more recent studies by Alfieri et al. (2015, 2017) and Winsemius et al. (2016).*

Thank you. We included those references in the revision.

P2 130-31: "Climate-change-induced" could be removed here, to avoid speculation.

Fixed.

P5 114-15: The sentence doesn't read well. Please reformulate.

Fixed.

*P5 l19: currently-frozen should be replaced with more appropriate terminology. Also, this sentence needs a supporting reference or a reason for the wider model spread.*

Fixed.

*P6 l28: also the over -> also over*

Fixed.

P7 l4: Is it available? Otherwise you should add "not shown"

We report the outcome of that analysis, which shows that final results are not sensitive to the order of averaging.

P8 114: flux to the Arctic Ocean

Fixed.

P8 126-27: "In the meantime" should be replaced with more appropriate terminology.

We changed the terminology to "also".

P11 15-7: This sentence sounds speculative as no specific simulation was performed to support it.

We show in figure 1 that streamflow is projected to increase in the regions near and above Arctic Circle and northern rivers, and in the 2nd paragraph of the Results section we argue that this results in increased fresh water flux into the Arctic Ocean, and we provide references to infer that : "*The projected increase in meltwater flux into the Arctic Ocean may contribute to sea level rise and changes in water salinity, temperature as well as circulation in the Arctic Ocean (Peterson et al., 2002; Rawlins et al., 2010)*". We have reported this as one of the general findings of the study in the Conclusion section.

Table 1: I suggest removing "rel" in the first two columns, as that is clear from the % sign.

Agreed. We removed the "rel." from the Table.

Figure 5 is surely the most interesting one, and the main novelty of this work. I wonder if the caption could be shortened. It is currently pretty long.

This is an important Figure, as you mentioned. It carries a lot of information which needs to be described to help readers better understand the results. However, we removed some sentences to shorten the caption.

**Anonymous Referee #3**

**Received and published: 19 July 2017**

This manuscript uses ISI-MIP streamflow simulations to explore the joint future of hydrological extremes (low and high flows). This is a quite relevant topic for HESS, but I have concerns about the novelty of the study, given the wealth of already published papers on hydrological extremes derived from ISI-MIP simulations. Furthermore, the statistical analysis is not (yet) convincing in my view – and choices are not justified (enough) – to bring the paper to a level where it could be published in HESS. All comments below have been initially drawn before I read comments from the two other referees, and I then added references to these in order to highlight common assessments or suggestions.

**Major Comments**

1. As mentioned above, and as already noted by Referee 1, there is little novelty in the topic and dataset used compared to previously published literature (especially to Giuntoli et al., 2015), and the little amount of novelty is not pushed forward in the manuscript. In my view, there are two new contributions: (1) the comparison between two contrasted RCPs, and (2) the quantification of absolute changes in high/low flow indices and their joint analysis. I agree with Referee 1 proposal to better highlight the manuscript's contributions, but I fear there are other issues that need to be tackled first.

Thank you. Accordingly, we have added more paragraphs to the introduction to emphasize on the knowledge gap that we are aiming to fill. To summarize a few, Giuntoli et al. 2015 studies changes in *frequency* of *un-routed* runoff (dimension  $[L.T^{-1}]$ ) extremes under *one* warming scenario, while we study changes in *magnitude/intensity* of *routed* runoff (streamflow, dimension  $[L^3.T^{-1}]$ ) extremes under *two* different warming scenarios. Un-routed runoff may not be directly used for flooding study, but the routed runoff (streamflow) has been used for this purpose in the literature instead (for instance: Koirala et al. 2014 and Hirabayashi et al. 2013). So although Giuntoli et al. 2015 is closest to ours in some respects, there are critical differences. Prudhomme et al 2014 also studies un-routed runoff, not streamflow. Koirala et al. 2014 studies 5th and 95th percentiles of streamflow, but routed by only one state-of-the-art GHM, and hence, does not account for GHM uncertainties. Hirabayashi et al. 2013 also uses a single state-of-the-art GHM,

to study the *frequency* of extremes. We study the change in the *magnitude/intensity* of the high and low extremes to understand the projected changes in the distribution of streamflow over the next century, while earlier studies (Giuntoli et al. 2015 for instance, and the others referred in the comments by reviewers) study changes in the *frequency* of streamflow passing the high and low extreme threshold. However, the thresholds used in those studies are defined based on the historical (20th century) distribution of streamflow. We find that the distribution of the streamflow is projected to change in the 21st century, and hence, the 95th/10th/5th percentiles used by these studies will not hold over the next century. Therefore, studying the changes in the *magnitude* of streamflow extremes can provide better interpretability of projected changes in the streamflow distribution compared to the *frequency*. Also, in the current work we study changes in flood and drought risks (both high and low flow extremes) together to understand the changes in streamflow distribution and simultaneous vulnerability profiles to different types of hydrological risk in different regions. We also study the changes under two emission scenarios (RCP2.6 and RCP8.5) to compare the impact of different levels of warming on streamflow distribution and consequently flood and drought risks (Giuntoli et al. 2015 uses only one (RCP8.5) warming scenario). We also calculate the number people affected by changes in different regions to understand whether the areas of increased risk coincide with highly populated regions.

2. The quantification of changes is, as already noted by the two other referees, first quite questionable in terms of wordings: high and low flow indices simply cannot be identified to flood and drought indices. Changes throughout the manuscript (including title) are therefore required. Furthermore, the authors consistently use the wording of streamflow in the manuscript, but I believe that the variable used is the (unrouted) runoff, as in previous related works on ISI-MIP data (Prudhomme et al, 2014; Giuntoli et al., 2015), and on the contrary to other works on (large) river basins (see e.g. Pechlivanidis et al., 2017; Vetter et al., 2017). This has serious implications for interpreting results in terms of floods and droughts (see the recent work by Zhao et al., 2017).

Giuntoli et al. 2015 studies changes in *frequency* of *un-routed* runoff (dimension  $[L.T^{-1}]$ ) extremes under *one* warming scenario, while we study changes in *magnitude/intensity* of *routed* runoff (streamflow, dimension  $[L^3.T^{-1}]$ ) extremes under *two* different warming scenarios. Unrouted runoff may not be directly used for flooding study, but the routed runoff (streamflow) has been used for this purpose in the literature instead (for instance: Koirala et al. 2014 and Hirabayashi et al. 2013). So although Giuntoli et al. 2015 is closest to ours in some respects, there are critical differences. Prudhomme et al 2014 also studies un-routed runoff, not streamflow.

In response to the comment made, we have omitted the words flood and drought from the revised title and use "streamflow extremes" instead. Accordingly, we refer to change in 95th and

5th percentiles of streamflow as change in flood and drought "*risk*", instead of direct change in flood and drought "*intensity*". We edited the manuscript accordingly.

3. The normalization procedure is probably interesting for positive variables like streamflow, as it makes multiplicative factors symmetrical with respect to zero. Multiplying (resp. dividing) present-day values by 3 results in a value of 1/2 (resp. -1/2). However, the lack of experience with dealing with such an index makes it rather difficult to interpret values. The way values converge towards 1 or -1 is for example not intuitive. The reader should at least be accompanied through this kind of basic examples.

Thank you for pointing this out. We added more explanations in the revision regarding how to interpret the normalized changes.

4. The joint analysis of changes in low flow and high flow indices is potentially attractive. However, I don't understand why the analysis is restricted to quadrants (cf. Figure 5) when all data are available for continuous assessments over the two indices (see Teuling et al., 2011a, b). This is in my view an oversimplification of the problem. You cannot identify with the quadrants a region with a small drought increase and a large flood increase (whatever that means). Moreover, the multimodel average is, as pointed out by Referee 1, potentially quite misleading. This is all the more problematic that there is a confusion (at least of the reader) when dealing with statistical significance. At several places in the manuscript, one may expect some tests for example on the sign of change within the multimodel ensemble (see the latest IPCC report), and not (only) the significance of changes between 30-year averages of future and present period for single models. Many detailed and interesting statistical analyses could be performed with this dataset by applying ANOVA techniques, and by for example deriving individual maps of GCMs/GHMs effects (in the ANOVA sense) on joint changes in low/high flow indices. This would avoid using latitude-averaged plots that do not convey in my view the most relevant information. For example, it is not possible on Figures 2, 3, and 4 to compare the spatial variance (along any given latitude) from the variance among GCMs/GHMs/combination of GCMs and GHMs (depending on the figure).

We do report the results for each of the indicators separately in Table 1 and Figure 1. We report mean value of the results in the abstract to summarize the global findings, and in the text to discuss the changes in a broader scale. However, we do provide the global maps of changes in each indicator, which show the magnitude of changes (represented by color saturation) in each grid cell and under each warming scenario. The scatter plots in Figure 2c&d also show the distribution of small and large changes under each scenario. We also show the results averaged by latitude to show the averaged changes over different climate boundaries (high latitudes, mid latitudes, tropics and subtropics, etc). We also provide the global maps for each of the models individually in the Supplementary Materials. We do agree that it may be more informative to add more calculations and techniques; that can be subject of another interesting study.

5. This also leads to my last major comment. I don't really understand why this study is restricted to only 5 GHMs. Statistical techniques are indeed available to take account of different sample sizes in ANOVA contexts (see for example Giuntoli et al., 2015). Furthermore, there is no justification in the manuscript on the choice of these specific 5 GHMs, and this has already been pointed out by Referee 1. This thus appears as a subjective and therefore negative choice for building confidence in results from this "ensemble of opportunity".

We removed the justification for the number of GHMs being same as the number of GCMs. We would not call the selected GHMs and GCMs only an "ensemble of opportunity" though; these models have had enough credibility to be selected to contribute to a project of the scale of the ISI-MIP, and have been used in earlier studies as well.

**Specific comments**

1. P1L16: percentage with respect to what period?

Over the 21st century, compared to the 20th century.

2. P2L14: Please make explicit what you mean by "impact". The hierarchy of impacts (for example in terms of monetary loss) is indeed highly dependent on the anthropogenic system under study.

Impact on the intensity. The beginning of the sentence is stating that increased amount of atmospheric water content is expected to intensify precipitation extremes, and the end of sentence is stating that the aforementioned impact (increase in intensity) is stronger for extreme precipitation than mean precipitation.

3. P2L19: I believe this is about "average runoff". Please specify.

Correct. Fixed.

4. P5L21: The normalization is announced and summarized here whereas it is described only much later on (P6 L3 ff.). Please reorganize the paragraphs.

We moved the paragraph to after definition of the metric.

5. P6L12-14: This is hardly understandable. Please consider giving the actual equations.

The conversion of relative change to normalized change is shown in Eq. 1. The conversion of normalized change to relative change however is done using numerical methods, as shown in the Supplementary Figure S1 (no equation available to report).

6. *P7L15-19*: *These figures are redundant with Table 1. Please rephrase.*

Rephrased.

7. P8L5, "with high agreement": Could you explain what you mean exactly here?

All models show large value of normalized increase in flood indicator and normalized decrease in drought indicator, as seen in Figure 2.

8. *P8L19*, *"fluctuations": Again, what do you mean here? Fluctuations in time, latitude, other? Please specify.*

Over latitude. Edited.

9. P8L23, "mean": over space, latitude? Please be more specific.

Mean over each latitudinal window. We edited the text to clarify.

10. P9L8, "statistically different": What is the test used here? Please be more specific on your statements.

It says "significantly different", not "statistically". The DBH model shows the opposite direction of change in drought indicator in the Northern Hemisphere. This is seen in the Figure 3 and the figure is referred in the text.

11. P10L11, "statistically significant": see above.

The determination of significance is based on a two-sample t-test. We say in the last paragraph of the Materials and Methods section that "*The two-sample t-test is used in this study to quantify the statistical significance level of difference between the means of the 20C and 21C streamflow time series*".

12. P11L16-17: This final sentence is rather ambiguous and wrongly suggests a 200

This comment seems to be incomplete. However, this sentence is correct, suggesting that the increase or decrease in flood and drought indicators is approximately twice as much in the RCP8.5 scenario as in the RCP2.6 scenario, as shown in Table 1 (compare the first two columns from the left).

**Technical corrections**

1. P2L3, "to be intensified": please rephrase

We rephrased it to "become more intense".

2. P2L8, "dictation": please rephrase

Rephrased.

3. P4L11: "increase"

Fixed.

*4. P6L17, "remained": please rephrase* Rephrased. 5. P8L14, missing "in" after "flux"

Fixed.

**Global change in flood and drought intensitiesstreamflow extremes under**

climate change in over the 21st century

3 Behzad Asadieh1,\*, Nir Y. Krakauer2

[1],[2] Civil Engineering Department and NOAA-CREST, The City College of New York, the City University of New York, New York, USA; basadie00@citymail.cuny.edu; nkrakauer@ccny.cuny.edu

\* Correspondence to Behzad Asadieh: basadie00@citymail.cuny.edu

9

10

1

2

4

5

6 7 8

**Abstract**

11 Global warming is expected to intensify the Earth's hydrological cycle and increase flood and drought risks. Changes over the 21st century under two warming scenarios in different 12 13 percentiles of the probability distribution of streamflow, and particularly of global high and low streamflow extremes (95th and 5th percentiles) over the 21st century under two warming 14 scenarios-are analyzed as indicators of hydrologic flood and drought intensity, using an 15 ensemble of bias-corrected global climate model (GCM) fields fed into different global 16 hydrological models (GHMs), to understand the changes in streamflow distribution and 17 18 simultaneous vulnerability to different types of hydrological risk in different regions-. Based on In the multi-model mean, under RCP8.5 scenario, approximately 37% of global land areas 19 20 experience increase in magnitude of extremely high streamflow (with an average increase of 24.5%), potentially increasing the chance of flooding in those regions. On the other hand, and 21 43% of global land areas are-show a exposed to increases in flood and drought 22 23 intensitiesdecrease in the magnitude of extremely low streamflow (average decrease of 24 51.5%), potentially increasing the chance of drought in those regions. respectively, by the end of the 21st century under RCP8.5 scenario. The average rates of increase in flood and drought 25 intensities in those areas are projected to be 24.5% and 51.5%, respectively.- Nearly About 26 27 10% of the global land areas are is under the potential projected risk of to face simultaneously 28 increase increasing- in both flood and drought intensities, with average rates of 10.1 and 19.8%, respectivelyhigh extreme streamflow and decreasing low extreme streamflow, 29 30 reflecting potentially worsening hazard of both flood and drought; further, these regions tend to be highly populated parts of the globe, currently holding around 30% of the world's 31 32 population (over 2.1 billion people). In a world more than 4 degrees warmer by the end of the Formatted: Not Superscript/ Subscript

[revised manuscript text omitted]

comparison with the end of the 20th century (1971-2000, 20C). We study changes in the

magnitude of the 95th percentile of annual streamflow (P95) in 21C compared to 20C, in which

an increase would be an indication of increase in the flood intensity. We also study the change in the magnitude of the 5th percentile (P5), in which a decrease would correspond to an increase

in the drought intensity. GHM-generated streamflow based on GCM inputs does not well capture the annual trends interannual variability in flow compared to observations, even where,

26

27

28

29 30

31 32

1 as in ISI-MIP, the GCM outputs are bias-corrected. However, the multi-decade average of 2 bias-corrected ISI-MIP streamflow is shown to be-more similar to that of observation-based 3 streamflow simulations (Asadieh et al., 2016). Other studies have also used relative changes in 4 multi-decade average of streamflow indices percentiles in a future 21C time window compared 5 to a historical 20C time window for flooding and streamflow extremes analyses (Dankers et al., 2013; Hirabayashi et al., 2013; Koirala et al., 2014; Tang and Lettenmaier., 2012). Alongside 6 7 the study of the magnitude of change, we also study the percentage of global population 8 affected by changes in high and low streamflow extremes, as an indication of the potential 9 impact of changes in flood and/or drought intensities events in those regions. Limiting global 10 warming to 2 degrees Celsius above the pre-industrial era (achievable in RCP2.6 scenario 11 (Moss et al., 2010; Stocker et al., 2013)) has been targeted in many scientific and governmental 12 plans, for instance the 2015 Paris Climate Agreement (UNFCCC, 2015). However, the 13 increasing trajectory of emissions observed over the beginning on the 21st century, if 14 continued, is more consistent with around 4 degrees Celsius of warming by the end of the 15 century (similar to RCP 8.5 scenario (Moss et al., 2010; Stocker et al., 2013)). Hence, we study both low and high radiative forcing scenarios (RCP2.6 and RCP8.5) to investigate the impacts 16 17 of 21C anthropogenic forcing on flood and drought risksstreamflow extremes.

**18 2. Materials and Methods**

19 We use daily streamflow data obtained from the first phase of the ISI-MIP (Warszawski et al., 20 2013). The ISI-MIP streamflow projections are produced by multiple GHMs, based on 21 bias-corrected meteorological outputs of 5 GCMs from the fifth version of the Coupled Model 22 Intercomparison Project (CMIP5) (Dankers et al., 2013), which are downscaled to 0.5 degree 23 resolution for the period 1971-2099. The GCMs contributing to the first phase of ISI-MIP are: 24 GFDL-ESM2M, HadGEM2-ES, IPSL-CM5A-LR, MIROC-ESM-CHEM and NorESM1-M 25 (Warszawski et al., 2013). The 5 GHMs selected for this study are WBM, MacPDM, 26 PCR-GLOBWB, DBH and LPJmL (refer to supplementary materials for details). These 27 models which have been used in previous studies, along with other models (Schewe et al., 28 2013). However, we limit the number of GHMs to 5 so the analysis in this global scale is 29 practical. ISI-MIP provides the streamflow outputs for only 5 GCMs, from more than 5 GHMs. 30 Here, the number of GHMs is also limited to 5 so that the uncertainties arising from the GCMs 31 and GHMs are readily comparable.

1 Relative changes in streamflow can be very large for individual grid cells, particularly in 2 eurrently-frozen high latitudes. This bias averages across models and grid cells toward a 3 sitive trend, as the decreases are limited to 100% loss of the historic flow, while the increase be well over 100% of the historic flow. Accordingly, changes are here normalized to 4 between 1 and +1, so the ranges of increases and decreases are comparable. Normalized 5 increase in the magnitude of P95 indicates increase in flood intensity, and is called the flood 6 7 indicator. Since increased drought intensity corresponds to decrease in the magnitude of P5, the 8 normalized changes in P5 are multiplied by 1 to form the drought indicator. We refer to 9 positive/negative change in flood (drought) indicator as increased/decreased flood (drought) 10 intensities, respectively.

IncreaseIncreasing-and-/decreasinge\_extreme in flood-high/and drought-low streamflow 11 can form four combinations, which are categorized as the following four quadrants: 1. 12 13 Increased flood high extreme and decreased low extreme and drought, 2. Increased flood high 14 and decreased droughtlow extreme, 3. Increased droughtDecreased high and decreased 15 flood low extreme, and 4. Decreased flood high extreme and drought 
[revised manuscript text omitted]

8 The shadings in the Figure 4 (inter-GHM uncertainty representative) is wider broader than 9 in the Figure 3 (inter-GCM uncertainty representative), which shows that the GHMs contribute 10 to higher rate of uncertainties in flood and droughtstreamflow change projections than GCMs. As seen in Figure 3 (c-d), for instance, drought-the P5 predictions of the DBH hydrological 11 model for Northern Hemisphere are significantly different from the other 4 hydrological 12 13 models considered here, even though the streamflow routings are based on the same GCM 14 inputs. Such inconsistency between DBH models and other models' results may not be 15 detectable, if, as the results are averaged as they are in the Figure 4, only the mean and standard deviation across GHMs is shown. Wide shadingsHigh uncertainties in Northern Hemisphere 16 17 drought low extreme trends in Figure 4 (c-d) shows high uncertaintiesreflects large disagreements among the GHMs for that region, while-the Figure 3 (c-d) reveals the major 18 19 cause of such uncertainties to be the DBH model.

Figure 5 illustrates the global maps of combined change in flood-high and drought 20 21 indicators low streamflow extremes under each RCP scenarios, obtained from the multi-model 22 mean results of-across all 25 GCM-GHM combination datasets. -Grid cells falling in each of the defined quadrants are shown with different colors, saturation of which is representative of 23 24 the intensity of changes. As shown in the Figure, northern high latitudes, especially north 25 Eurasia, northern Canada and Alaska, as well as eastern Africa and parts of South and 26 Southeast Asia and Eastern Oceania show increase in flood intensity the magnitude of high 27 streamflow extremes (P95) in both scenarios, similar to findings of earlier studies and reflecting a potential for increasing flood hazard (Dankers et al., 2013; Hirabayashi et al., 2013; 28 29 Schewe et al., 2013). Central America, Southern Africa, Middle East, Southern Europe, 30 Mediterranean and major parts of South America and Australia show increase in drought 31 intensity decrease in the magnitude of low streamflow extrem (P5) in both scenarios, comparable to findings of earlier studies and reflecting a potential for increasing drought 32 33 hazard (Arnell, 2004; Dai, 2011; Hagemann et al., 2013; Schewe et al., 2013). The United

1 Kingdom and the shores of the North Sea as well as large parts of Tibetan, South Asia and 2 Western Oceania show increase in potential for both flood and drought intensitieshazards 3 (increase in P95 and decrease in P5). In these cases, while preserving the direction of change, 4 the RCP8.5 scenario projects stronger magnitude change compared to the RCP2.6 scenario. 5 Southern and Western Europe and southern parts of the United States show small-magnitude, mixed-sign increases in flood and drought changes in P95 and P5intensities in the RCP2.6 6 7 scenario. However, projections under RCP8.5 scenario are for strong increase in 8 droughtdecrease in P5 in those regions, suggesting increasing potential for drought hazard. 9 Some parts of eastern Russia and northern United States show decreases in P95 and increases in P5, suggesting the potential for reduction in both flood and drought hazardsintensities 10 11 (Figure 5). Compared to RCP2.6, the RCP8.5 scenario shows more expansion in drought 12 intensity and less expansion in flood intensity (Figure 5 and Table 2).

13 Under the low radiative forcing scenario (RCP2.6), 45.4% of global land area shows 14 increase in flood intensity high extreme in the multi-model mean and 36.4% shows increase in 15 droughtdecrease in low extreme, indicating more land area exposure exposed to increasing flood intensity hazard compared to than to drought hazard. The high radiative forcing scenario 16 (RCP8.5) projections show the opposite outcome, with projects increased flood intensity high 17 extreme streamflow in 36.6% of global land area and increased droughtdecreased low extreme 18 19 in 43.2%. Unlike the RCP2.6 scenario, the RCP8.5 scenario projects more land area exposure 20 exposed to increasing drought intensity hazard comparedthan to flood. Moreover, flood and 21 drought eventschanges in streamflow extremes are more intenselarger 
[revised manuscript text omitted]
)-                                           | <del>rel.</del> )                                                                                              | <del>rel.</del> )               |                                                                                       |         |                                                                             |         |   | Formetted Fort Italia Conseler Societ Fo                     |
| High extreme (P95)                                                     |                                                                                                                |                                 |                                                                                       | -       |                                                                             |         | - | Italic                                                       |
| Decreased cells                                                        | -0.1178                                                                                                        | -0.0539                         |                                                                                       |         |                                                                             |         |   | Formatted: Font: Italic, Complex Script Fo                   |
| ( Decreased flood
potential) Flood Decreased
Cells | (-21.10 %-
<del>rel.</del> )                                                                                | (-10.25 %-
<del>rel.</del> ) | 39.2%                                                                                 | 30.5%   | 42.2%                                                                       | 32.2%   |   | Formatted: Font: Italic, Complex Script For
Italic |
| Low extreme (P5)                                                       |                                                                                                                |                                 |                                                                                       | _       |                                                                             | -       | - |                                                              |
| Decreased cells                                                        | - 0.2045                                                                                                | - 0.1029                 |                                                                                       |         |                                                                             |         |   |                                                              |
| (Increased drought                                                     | (- 51.40 %-                                                                                             | ( - 22.95 %-             | 43.2%                                                                                 | 36.3%   | 67.8%                                                                       | 56.1%   |   | Formatted: Font: Italic, Complex Script Fo                   |
| potential) Drought Increased
Cells                           | <del>rel.</del> )                                                                                              | <del>rel.</del> )               |                                                                                       |         |                                                                             |         |   | Italic                                                       |
| Low extreme (P5)                                                       |                                                                                                                |                                 |                                                                                       |         |                                                                             |         | - |                                                              |
| Increased cells                                                        | -0.1784                                                                                                        | -0.1018                         |                                                                                       |         |                                                                             |         |   |                                                              |
| (Decreased drought                                                     | (-30.30 %-                                                                                                     | (-18.50 %-                      | 32.7%                                                                                 | 39.6%   | 28.1%                                                                       | 39.8%   |   |                                                              |
| potential)Drought Decreased                                            | rel.)                                                                                                          | rel.)                           |                                                                                       |         |                                                                             |         |   |                                                              |
| Cells                                                                  | - /                                                                                                            |                                 |                                                                                       |         |                                                                             |         |   |                                                              |
|                                                                        |                                                                                                                |                                 |                                                                                       |         |                                                                             |         |   |                                                              |

Table 2. Percent of population and land area affected by each flood and droughthigh and low extreme change quadrants, for RCP2.6 and RCP8.5 scenarios. Presented percentages are for total global land area and total global population. Hence, the percentages presented for quads. 1-4 sum up to the 75.9% of global land area and 95.9% of the year 2015 total global population considered in this study.

|                                         |        | Quad. 1. increased
high extreme and
decreased low
extreme flood and
<del>drought increased</del> | Quad. 2.
increased high
and low
extremeflood-
<del>increased,</del>
<del>drought-</del>
<del>decreased</del> | Quad. 3.
decreased high
and low
extremedrought
increased, flood-
<del>decreased</del> | Quad. 4. decreased
high extreme and
increased low
extremeflood and
<del>drought decreased</del> |
|-----------------------------------------|--------|------------------------------------------------------------------------------------------------------------------------------------------|-----------------------------------------------------------------------------------------------------------------------------------------------------|---------------------------------------------------------------------------------------------------------------------------|-----------------------------------------------------------------------------------------------------------------------------------------|
| Land area affected (% of total 148.9    | RCP8.5 | 9.6%                                                                                                                                     | 27.0%                                                                                                                                               | 33.6%                                                                                                                     | 5.7%                                                                                                                                    |
| million km 2 )               | RCP2.6 | 10.8%                                                                                                                                    | 34.5%                                                                                                                                               | 25.5%                                                                                                                     | 5.1%                                                                                                                                    |
| Population affected
(% of total 7.13 | RCP8.5 | 29.6%                                                                                                                                    | 24.1%                                                                                                                                               | 38.2%                                                                                                                     | 4.0%                                                                                                                                    |
| billion people)                         | RCP2.6 | 27.1%                                                                                                                                    | 35.6%                                                                                                                                               | 28.9%                                                                                                                     | 4.3%                                                                                                                                    |

Formatted Table

Table 3. Multi-model average change in flood and drought indicatorshigh and low streamflow extremes, averaged for each quadrant, for RCP2.6 and RCP8.5 scenarios. The numbers show the normalized change and the numbers in parenthesis show the changes reverted to the relative percentages.

|        | Quad. 1. increased high
extreme and decreased
low extremeQuad. 1.
flood and drought
increased |                                                                                   | Quad. 2. inc
and low extr
flood increa
decre             | creased high
emeQuad. 2.
sed, drought
sased                                         | Quad. 3. dee
and low extr
drought incr
decre                         | creased high
emeQuad. 3.
reased, flood
cased                                       | Quad. 4. decreased high
extreme and increased
low extremeQuad. 4.
flood and drought
decreased |                                                                                             |
|--------|-----------------------------------------------------------------------------------------------------------|-----------------------------------------------------------------------------------|-------------------------------------------------------------------|----------------------------------------------------------------------------------------------|-------------------------------------------------------------------------------|---------------------------------------------------------------------------------------------|-----------------------------------------------------------------------------------------------------------|---------------------------------------------------------------------------------------------|
|        | Change in
<del>floodhigh
ext.</del>                                                         | Change in
low
ext.Chang
<del>e in-</del>
drought | Change in
high
ext.Chang
e in flood | Change in
low
ext.Chang
<del>e in-</del>
<del>drought</del> | Change in
high
ext.<del>Chang</del>
e in flood | Change in
low
ext.Chang
<del>e in</del>
<del>drought</del> | Change in
high
ext.<del>Chang</del>
e in flood                             | Change in
low
ext.Chang
<del>e in</del>
<del>drought</del> |
| RCP8.5 | 0.0481
(10.10 %)                                                                                       | - 0.0901
( - 19.80 %)                                            | 0.1311
(30.20 %)                                               | -0.1909
(-32.05 %)                                                                        | -0.1290
(-22.85 %)                                                         | - 0.2372
( - 62.20 %)                                                      | -0.0508
(-9.65 %)                                                                                      | -0.1183
(-21.15 %)                                                                       |
| RCP2.6 | 0.0306                                                                                                    | 0.0556
( - 11.80 %)                                                     | 0.0700
(15.05 %)                                               | -0.1074
(-19.40 %)                                                                        | -0.0593
(-11.20 %)                                                         | - 0.1230
( - 28.05 %)                                                      | -0.0267
(-5.20 %)                                                                                      | -0.0635
(-11.95 %)                                                                       |